# Locally Invariant Explanations: Towards Causal Explanations through Local Invariant Learning

## Abstract

Locally interpretable model agnostic explanations (LIME) method is one of the most popular methods used to explain black-box models at a per example level. Although many variants have been proposed, few provide a simple way to produce high fidelity explanations that are also stable and intuitive. In this work, we provide a novel perspective by proposing a model agnostic local explanation method inspired by the invariant risk minimization (IRM) principle – originally proposed for (global) out-of-distribution generalization – to provide such high fidelity explanations that are also stable and unidirectional across nearby examples. Our method is based on a game theoretic formulation where we theoretically show that our approach has a strong tendency to eliminate features where the gradient of the black-box function abruptly changes sign in the locality of the example we want to explain, while in other cases it is more careful and will choose a more conservative (feature) attribution, a behavior which can be highly desirable for recourse. Empirically, we show on tabular, image and text data that the quality of our explanations with neighborhoods formed using random perturbations are much better than LIME and in some cases even comparable to other methods that use realistic neighbors sampled from the data manifold, where the latter is a popular strategy to obtain high quality explanations. This is a desirable property given that learning a manifold to either create realistic neighbors or to project explanations is typically expensive or may even be impossible. Moreover, our algorithm is simple and efficient to train, and can ascertain stable input features for local decisions of a black-box without access to side information such as a (partial) causal graph as has been seen in some recent works.

## 1 Introduction

Deployment and usage of neural black-box models has significantly grown in industry over the last few years creating the need for new tools to help users understand and trust models (Gunning, 2017). Even well-studied application domains such as image recognition require some form of prediction understanding in order for the user to incorporate the model into any important decisions (Simonyan et al., 2013; Lapuschkin et al., 2016). An example of this could be a doctor who is given a cancer diagnosis based on an image scan. Since the doctor holds responsibility for the final diagnosis, the model must provide sufficient reason for its prediction. Even new text categorization tasks (Feng et al., 2018) are becoming important with the growing need for social media companies to provide better monitoring of public content. Twitter recently began monitoring tweets related to COVID-19 in order to label tweets containing misleading information, disputed claims, or unverified claims (Roth & Pickles, 2020). Laws will likely emerge requiring explanations for why red flags were or were not raised in many examples. In fact, the General Data Protection and Regulation (GDPR) (Yannella & Kagan, 2018) act passed in Europe already requires automated systems that make decisions affecting humans to be able to explain them. Given this acute need, a number of methods have been proposed to explain local decisions (i.e. example specific decisions) of classifiers (Ribeiro et al., 2016; Lundberg & Lee, 2017; Simonyan et al., 2013; Lapuschkin et al., 2016; Dhurandhar et al., 2018a). Locally interpretable model-agnostic explanations (LIME) is arguably the most well-known local explanation method that requires only query (or black-box) access to the model.

Although LIME is a popular method, it is known to be sensitive to certain design choices such as i) (random) sampling to create the *(perturbation) neighborhood*[1], ii) the size of this neighborhood (number of samples) and iii) (local) fitting procedure to learn the explanation model (Molnar, 2019; Zhang et al., 2019b). The first, most serious issue could lead to nearby examples having drastically different explanations making effective recourse a challenge. One possible mitigation is to increase the neighborhood size, however, one cannot arbitrarily do so as it not only leads to higher computational cost, but in today's cloud computing-driven world it could have direct monetary implications where every query to a black-box model has an associated cost (Dhurandhar et al., 2019).

There have been variants suggested to overcome some of these limitations (Botari et al., 2020; Shrotri et al., 2021; Plumb et al., 2018) primarily through mechanisms that create realistic neighborhoods or through adversarial training (Lakkaraju et al., 2020), however, their efficacy is restricted to certain settings and modalities based on their assumptions and training strategies.

In this paper we introduce a new method called Locally INvariant EXplanations (LINEX) inspired by the invariant risk minimization (IRM) principle (Arjovsky et al., 2019), that produces feature based explanations that are robust to neighborhood sampling and can recover faithful (i.e. mimic black-box behavior), stable (i.e. similar for closeby examples) and unidirectional (i.e. same sign attributions for closeby examples, see section 4.1) explanations across tabular, image, and text modalities. In particular, we show that our method performs better than the competitors for random as well as realistic neighborhood generation, where in some cases even with the prior strategy our explanation quality is close to methods that employ the latter. Qualitatively, our method highlights (local) features as important that in the particular locality i) have consistently high gradient with respect to (w.r.t.) the black-box function and ii) where the gradient does not change significantly, especially in sign. Such stable behavior for LINEX is visualized in Figure 1, where we get similar explanations for nearby examples in the IRIS dataset. The (in)fidelity of LINEX is still similar to LIME (see Table 3), but of course our explanations are much more stable.

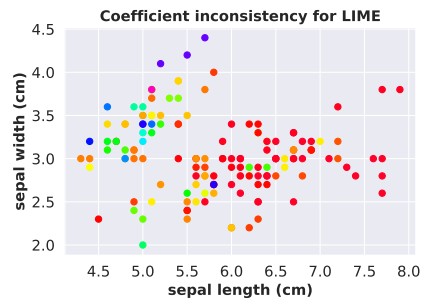

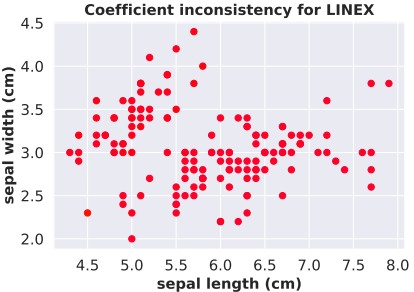

Figure 1: Above we visualize for the IRIS dataset the Coefficient Inconsistency (CI) (see Section 5 for exact definition and setup details) between the explanation (top two features) for an example and its nearest neighbor in the dataset. Each circle denotes an example and a *rainbow* colormap depicts the degree of inconsistency w.r.t. its nearest neighbor where red implies least inconsistency, while violet implies the most. As can be seen LINEX explanations are much more consistent than LIME's.

## 2 RELATED WORK

Posthoc explanations can typically be partitioned into two broad categories global and local. Global explainability avers to trying to understand a black-box model at a holistic level where the typical tact is knowledge transfer (Hinton et al., 2015; Dhurandhar et al., 2018b; 2020) where (soft/hard) labels of the black-box model are used to train an interpretable model such as a decision tree or rule list (Rudin, 2019). Local explanations on the other hand avers to understanding individual decisions. These explanations are typically in two forms, either exemplar based or feature based. For exemplar based as the name suggests similar but diverse examples (Kim et al., 2016; Gurumoorthy et al., 2019) are provided as explanations for the input in question. While for feature based (Ribeiro et al., 2016; Lundberg & Lee, 2017; Dhurandhar et al., 2018a; Lapuschkin et al., 2016), which is the focus of this work, important features are returned as being important for the decision made for the input. There are some methods that do both (Plumb et al., 2018). Moreover, there are methods which provide explanations that are local, global as well as at a group level (Ramamurthy et al., 2020). All of these methods though may not still provide stable and robust local feature based explanations which can be desirable in practice (Ghorbani et al., 2019).

---

[1]By *perturbation neighborhood* – referred to as simply *neighborhood* – we mean neighborhoods generated for local explanations. By *exemplar neighborhood*, we mean nearest examples in a dataset to a given example.

Given this there have been more recent works that try to learn either robust or even causal explanations. In (Lakkaraju et al., 2020) the authors try to learn robust and stable local explanations relative to distribution shifts and adversarial attacks. However, the distribution shifts they consider are linear shifts and adversarial training is performed which can be slow and sometimes unstable (Zhang et al., 2019a). Moreover, the method seems to be applicable primarily to tabular data. Works on causal explanations (Frye et al., 2020; Heskes et al., 2020) mainly modify SHAP and assume access to a partial causal graph. Some others (Vig et al., 2020) assume white-box access. In this work we do not assume availability of such additional information. There are also works which show that creating realistic neighborhoods by learning the data manifold for LIME (Botari et al., 2020; Shrotri et al., 2021) can lead to better quality explanations, where in a particular work (Anders et al., 2020) it is suggested that projecting explanations themselves on to the manifold can also make them more robust. The need for stability in a exemplar neighborhood for LIME like methods has been highlighted in (Zhang et al., 2019b), with the general desire for stable explanations being also expressed in prior works (Yeh et al., 2019; Visani et al., 2020).

## 3 PRELIMINARIES

**Invariant Risk Minimization:** Given a collection of training datasets $D = \{D_e\}_{e \in \mathcal{E}_{tr}}$ gathered from a set of environments $\mathcal{E}_{tr}$, where $D_e = \{\boldsymbol{x}_e^i, y_e^i\}_{i=1}^{n_e}$ is the dataset gathered from environment $e \in \mathcal{E}_{tr}$ and $n_e$ is the number of points in environment $e$. The feature value for data point $i$ is $\boldsymbol{x}_e^i \in \mathcal{X}$ and the corresponding label is $y_e^i \in \mathcal{Y}$, where $\mathcal{X} \subseteq \mathbb{R}^d$ and $\mathcal{Y} \subseteq \mathbb{R}$. Each point $(\boldsymbol{x}_e^i, y_e^i)$ in environment $e$ is drawn i.i.d from a distribution $\mathbb{P}_e$. Define a predictor $f : \mathcal{X} \rightarrow \mathbb{R}$. The goal of IRM is to use these collection of datasets $D$ to construct a predictor $f$ that performs well across many unseen environments $\mathcal{E}_{all}$, where $\mathcal{E}_{all} \supseteq \mathcal{E}_{tr}$. Define the risk achieved by $f$ in environment $e$ as $R_e(f) = \mathbb{E}_e\big[\ell(f(\boldsymbol{X}_e), Y_e)\big]$, where $\ell$ is the square loss when $f(\boldsymbol{X}_e)$ is the predicted value and $Y_e$ is the corresponding label, $(\boldsymbol{X}_e, Y_e) \sim \mathbb{P}_e$ and the expectation $\mathbb{E}_e$ is defined w.r.t. the distribution of points in environment $e$.

An invariant predictor is composed of two parts a representation $\boldsymbol{\Phi} \in \mathbb{R}^{d \times n}$ and a predictor (with the constant term) $\boldsymbol{w} \in \mathbb{R}^{d \times 1}$. We say that a data representation $\boldsymbol{\Phi}$ elicits an invariant predictor $\boldsymbol{w}^\mathsf{T}\boldsymbol{\Phi}$ across the set of environments $\mathcal{E}_{tr}$ if there is a predictor $\boldsymbol{w}$ that achieves the minimum risk for all the environments $\boldsymbol{w} \in \arg\min_{\tilde{\boldsymbol{w}} \in \mathbb{R}^{d \times 1}} R_e(\tilde{\boldsymbol{w}}^\mathsf{T}\boldsymbol{\Phi})$, $\forall e \in \mathcal{E}_{tr}$. IRM may be phrased as the following constrained optimization problem (Arjovsky et al., 2019):

$$\min_{\boldsymbol{\Phi} \in \mathbb{R}^{d \times n}, \boldsymbol{w} \in \mathbb{R}^{d \times 1}} \sum_{e \in \mathcal{E}_{tr}} R_e(\boldsymbol{w}^\mathsf{T}\boldsymbol{\Phi}) \quad \text{s.t. } \boldsymbol{w} \in \arg\min_{\tilde{\boldsymbol{w}} \in \mathbb{R}^{d \times 1}} R_e(\tilde{\boldsymbol{w}}^\mathsf{T}\boldsymbol{\Phi}), \ \forall e \in \mathcal{E}_{tr} \tag{1}$$

If $\boldsymbol{w}^\mathsf{T}\boldsymbol{\Phi}$ solves the above, then it is an invariant predictor across the training environments $\mathcal{E}_{tr}$.

**Local explainability setup vs IRM setup:** There are multiple differences between IRM setup and the setup we have in this paper. First, IRM is meant to learn global models directly from the dataset. This also applies to the game theoretic version proposed in (Ahuja et al., 2020), while for local explainability we are trying to learn local models that explain (per example) a given black-box model. Second, given that we want to explain a black-box model which typically is a function of the input features, we want to highlight features in our explanation that may be spurious from the domain perspective, but nonetheless the black-box model uses them to make decisions. Third, the IRM model does not have to be interpretable as its learned representation $\boldsymbol{\Phi}$ can be arbitrary. Fourth, the IRM learning procedure is quite inefficient as it tries to solve a bi-level optimization problem with a highly non-convex constraint. Fifth, IRM assumes that one has access to multiple environments (viz. multiple data sources), but in our case we have to figure out how to appropriately create them for each example that we want to explain. Last, since a black-box model is trained on the input features all or a subset of them are causal to its prediction, unlike the standard IRM setup where a predictor is expected to model the true underlying mechanism. Hence in our case, it is sufficient to limit ourselves to some interpretable representation, which in many cases is just the input feature space (i.e. $\boldsymbol{\Phi} = \mathcal{X}$).

Thus IRM cannot be directly applied to our problem and we propose a novel game theoretic approach that is suitable for local explainability consistent with the above points. With our approach it is interesting that we are able to obtain similar style theoretical results for our setting that have been seen in out-of-distribution generalization type of works (Ahuja et al., 2021), thus justifying our choice of leveraging such frameworks.

**Nash Equilibrium (NE):** To understand how certain key aspects of our method function let us revisit the notion of Nash Equilibrium (Dutta, 1999). A standard normal form game is written as a tuple $\Omega = (\mathcal{N}, \{u_i\}_{i \in \mathcal{N}}, \{\mathcal{S}_i\}_{i \in \mathcal{N}})$, where $\mathcal{N}$ is a finite set of players. Player $i \in \mathcal{N}$ takes actions from a strategy set $\mathcal{S}_i$. The utility of player $i$ is $u_i : \mathcal{S} \to \mathbb{R}$, where we write the joint set of actions of all the players as $\mathcal{S} = \Pi_{i \in \mathcal{N}} \mathcal{S}_i$. The joint strategy of all the players is given as $s \in \mathcal{S}$, the strategy of player $i$ is $s_i$ and the strategy of the rest of players is $s_{-i} = (s_{i'})_{i' \neq i}$.

**Definition 1.** *A strategy $s^{\dagger} \in \mathcal{S}$ is said to be a pure strategy Nash equilibrium (NE) if it satisfies*

$$u_i(s_i^{\dagger}, s_{-i}^{\dagger}) \geq u_i(k, s_{-i}^{\dagger}), \forall k \in \mathcal{S}_i, \forall i \in \mathcal{N}$$

NE thus identifies a state where each player is using the best possible strategy in response to the rest of the players leaving no incentive for any player to alter their strategy. In seminal work by (Debreu, 1952) it was shown that for a special class of games called concave games such a pure NE always exists. This is relevant because the game implied by Algorithm 1 falls in this category.

## 4 METHODOLOGY

We first define certain desirable properties we would like our explanation method to have. The first three have been seen in previous works, while the last *Unidirectionality* is something new we propose. We then describe our method where the goal is to explain a black-box model $f : \mathcal{X} \to \mathbb{R}$ for individual inputs $x$ based on predictors $w$ by looking at their corresponding components, also termed as feature attributions.

### 4.1 DESIRABLE PROPERTIES

We now discuss certain properties we would like our explainability method to have in order to provide robust explanations that could potentially be used for recourse.

**Fidelity:** This is the most standard property which all proxy model based explanation methods are evaluated against (Ribeiro et al., 2016; Lundberg & Lee, 2017; Lakkaraju et al., 2020) as it measures how well the proxy model simulates the behavior of the black-box it is attempting to explain.

**Stability:** This is also a popular notion (Hancox-Li, 2020; Ramamurthy et al., 2020; Yeh et al., 2019) to evaluate robustness of explanations. Largely, stability can be measured at three levels. One is prediction stability, which measures how much the predictions of an explanation model change for the same example subject to different randomizations within the method or across close by examples. The second is the variance in the feature attributions again for the same or close by examples. It is good for a method to showcase stability w.r.t. both even though in many cases the latter might imply the former. An interesting third notion of stability is the correlation between the feature attributions of an explanation model and average feature values of examples belonging to a particular class. This measures how much does the explanation method pick features that are important for the class, rather than spurious ones that seem important for the example of interest.

**Black-box Invariance:** This is the same as implementation invariance defined in (Sundararajan et al., 2017). Essentially, if two models have exactly the same behavior on all inputs then their explanations should also be the same. Since, our method is model agnostic with only query access to the model it is easy to see that it satisfies this property if the same environments are created.

**Unidirectionality:** This is a new property, but as we argue that this is a natural one to have. Loosely speaking, unidirectionality would measure how consistently the sign of the predictor for a feature is maintained for the same or close by examples by an explanation method. This is a natural metric (Miller, 2018), which from an algorithmic recourse (Karimi et al., 2021) perspective is also highly desirable. For instance, recommending a person to increase their salary to get a loan and then recommending to another person with a very similar profile to decrease their salary for the same outcome makes little sense.

We define the unidirectionality $\Upsilon$ as a measure of how consistent the sign of the attribution for a particular feature in a local explanation is when varying neighborhoods for the same example or when considering different close by examples. In particular, given $m$ attributions for each of $d$ features

---

**Algorithm 1:** Locally Invariant EXplanations (LINEX) method.

---

**Input:** example to explain $\boldsymbol{x}$, black-box predictor $f(.)$, number of environments to be created $k$, $(l_\infty)$ threshold $\gamma > 0$, $(l_1)$ threshold $t > 0$ and convergence threshold $\epsilon > 0$

**Initialize:** $\forall i \in \{1, ..., k\}$ $\tilde{\boldsymbol{w}}_i = \boldsymbol{0}$ and $\Delta = 0$

Let $\xi_1(.), ..., \xi_k(.)$ be $k$ environment creation functions as described in section 4.2.2

**do**

    **for** $i = 1$ *to* $k$ **do**

        $\tilde{\boldsymbol{w}}_{-i}^+ = \sum_{j \in \{1,...,k\}, j \neq i} \tilde{\boldsymbol{w}}_j$

        $\tilde{\boldsymbol{w}}_i^{\mathsf{prev}} = \tilde{\boldsymbol{w}}_i$

        $\tilde{\boldsymbol{w}}_i = \arg\min_{\tilde{\boldsymbol{w}}} \sum_{\tilde{\boldsymbol{x}} \in \xi_i(\boldsymbol{x})} \left( f(\tilde{\boldsymbol{x}}) - \tilde{\boldsymbol{w}}_{-i}^{+\mathsf{T}} \tilde{\boldsymbol{x}} - \tilde{\boldsymbol{w}}^{\mathsf{T}} \tilde{\boldsymbol{x}} \right)^2$ s.t. $|\tilde{\boldsymbol{w}}_{-i}^+ + \tilde{\boldsymbol{w}}|_1 \leq t$ and $|\tilde{\boldsymbol{w}}|_\infty \leq \gamma$

        $\Delta = \max\left(|\tilde{\boldsymbol{w}}_i^{\mathsf{prev}} - \tilde{\boldsymbol{w}}_i|_2, \Delta\right)$

    **end**

**while** $\Delta \geq \epsilon$;

**Output:** $\boldsymbol{w} = \sum_{i \in \{1,...,k\}} \tilde{\boldsymbol{w}}_i$

---

denoted by $w_1^{(1)}, ..., w_m^{(d)}$ the unidirectionality metric for an example is:

$$\Upsilon = \frac{1}{md} \sum_{i=1}^{d} \left| \sum_{j=1}^{m} \mathsf{sgn}\left(w_j^{(i)}\right) 1 \right| \tag{2}$$

where $|.|$ stands for absolute value. Clearly, the more consistent the signs for the attribution of a particular feature across $m$ realizations/explanations the higher the value, where the maximum value can be one. If equal number of attributions have different signs for all features then $\Upsilon$ will be zero, the lowest possible value. This property thus measures how intuitively consistent (ignoring magnitude) the explanations are and compliments the other properties mentioned above.

## 4.2 METHOD

### 4.2.1 DESCRIPTION

In Algorithm 1, we show the steps of our method LINEX. The input to the method is the example we want to explain $\boldsymbol{x}$, the black-box predictor, a few thresholds that we describe next and $k$ (local) environments whose creation is described in Section 4.2.2.

In the algorithm we iteratively learn a constrained least squares predictor for each environment, where the final (local) linear predictor is the sum of these individual predictors. In each iteration when computing the contribution of environment $e_i$ to the final summed predictor, the most recent contributions of the other predictors are summed and the residual is optimized subject to the constraints. The first constraint is a standard lasso type constraint which tries to keep the final predictor sparse as is also seen in LIME.

**Why $l_\infty$ constraint?** The second constraint is more unique and is a $l_\infty$ constraint on the predictor of just the current environment. This constraint as we prove in Section 4.3 is essential for obtaining robust predictors. To intuitively understand why this is the case consider we have two environments. In this case if the optimal predictors for a feature in each environment have opposite signs, then the Nash equilibrium (NE) is when each predictor takes $+\gamma$ or $-\gamma$ values as they try to force the sum to have the same sign as them. *In other words, features that have a disagreement in even the direction of their impact are eliminated by our method.* LIME type methods on the other hand would simply choose some form of average value of the predictors which may be a risky choice especially for actionability/recourse given that the directions change so abruptly. On the other hand, if the optimal predictors for a feature in the two environments have the same sign, the lower absolute valued predictor would be chosen (assuming $\gamma$ is greater) making it a careful choice. The reasoning for this and a discussion involving more than two environments is given in Section 4.3.

The overall algorithm resembles a (simultaneous) game where each environment is a player trying to find the best predictor for its environment given all the other predictors and constraints. Also note that the optimization problem solved by each player is convex as norms are convex.

### 4.2.2 CREATING LOCAL ENVIRONMENTS

In the standard IRM framework environments are assumed to be given, however, in our case of local explainability we have to decide how to produce them. We offer a few options for the environment creation functions $\xi_i \; \forall i \{1, ..., k\}$ in Algorithm 1.

**Random Perturbation:** This possibly is the simplest approach and similar to what LIME employs. We could perturb the input example by adding zero mean gaussian noise to create the base environment (used by LIME) and then perform bootstrap sampling to create the $k$ different environments. This will efficiently create neighbors in each environment, although they may be unrealistic in the sense that they could correspond to low probability points w.r.t. the underlying distribution.

**Realistic Generation/Selection:** One could also create neighbors using data generators such as done in MeLIME (Botari et al., 2020) or select neighboring examples from the training set as done in MAPLE (Plumb et al., 2018) to create the base environment following which bootstrap sampling could be done to form the $k$ different environments. This approach may provide more realistic neighbors than the previous one, but may be much more computationally expensive.

Other than bootstrapping one could also over sample and try to find the optimal hard/soft partition through various clustering type objectives (Aggarwal & Reddy, 2013; Creager et al., 2020).

## 4.3 THEORETICAL RESULTS

In this section, we analyze the output of Algorithm 1 when there are two environments. The extension to multiple environments is discussed following this result, where the general intuition is still maintained but some special cases arise depending on whether there are an even or odd number of environments. To prove our main result we make two assumptions.

**Assumption 1** *The feature values for each of the dimensions in the samples created forming the local environments are independent.*

This assumption is satisfied by the most standard way of creating neighborhoods/environments, where random gaussian noise is used to create them as described in Section 4.2.2.

**Assumption 2** $t \geq \gamma d$, *where $d$ is the dimensionality of the feature vector.*

Making this assumption ensures that we closely analyze the role of the $\ell_\infty$ penalty, which is one of the main novelties in our method.

**Definition 2** Let the explanation that each environment $\xi_i$ arrives at for an example $\boldsymbol{x}$ based on unconstrained least squares minimization be $\boldsymbol{w}_i^*$ where,

$$\boldsymbol{w}_i^* \in \underset{\tilde{w} \in \mathbb{R}^d}{\arg\min} \; \mathbb{E}_{\tilde{\boldsymbol{x}} \in \xi_i(x)}[(f(\tilde{\boldsymbol{x}})^2 - \tilde{w}^\top \tilde{\boldsymbol{x}})^2] \tag{3}$$

The expectation is taken w.r.t the environment generation distribution.

**Theorem 1.** *The output of Algorithm 1 under Assumptions 1, 2 and equation 3 is given by:*

$$\boldsymbol{w} = \left( \boldsymbol{w}_1^* \odot \mathbf{1}_{|\boldsymbol{w}_2^*| \geq |\boldsymbol{w}_1^*|} + \boldsymbol{w}_2^* \odot \mathbf{1}_{|\boldsymbol{w}_1^*| > |\boldsymbol{w}_2^*|} \right) \mathbf{1}_{\boldsymbol{w}_1^* \odot \boldsymbol{w}_2^* \geq 0} \tag{4}$$

*where $\odot$ is element wise product and $\mathbf{1}$ is the indicator function.*

*Proof Sketch.* The above expression describes the NE of the game played between the two local environments each trying to move $\boldsymbol{w}$ towards their least squares optimal solution. Given assumptions 1 and 2, we witness the following behavior of our method. Let the $i^{th}$ feature of the predictors $\tilde{w}_1$ and $\tilde{w}_2$ from Algorithm 1 be $\tilde{w}_{1i}$ and $\tilde{w}_{2i}$ respectively. Let the corresponding least squares optimal predictors for the $i^{th}$ feature have the following relation: $w_{1i}^* > w_{2i}^*$ and $|w_{1i}^*| > |w_{2i}^*|$. Then the two environments will push the ensemble predictor, $\tilde{w}_{1i} + \tilde{w}_{2i}$, in opposite directions during their turns, with the first environment increasing its weight, $\tilde{w}_{1i}$, and the second environment decreasing its weight, $\tilde{w}_{2i}$. Eventually, the environment with a higher absolute value ($\xi_1 = 1$ since $|w_{1i}^*| > |w_{2i}^*|$) reaches the boundary ($\tilde{w}_{1i} = \gamma$) and cannot move any further due to the $l_\infty$ constraint. The other environment $\xi_2$ best responds, where it either hits the other end of the boundary ($\tilde{w}_{2i} = -\gamma$), in which case the weight of the ensemble for component $i$ is zero, a case which occurs if $w_{1i}^*$ and $w_{2i}^*$ have opposite signs; or gets close to the other boundary while staying in the interior ($\tilde{w}_{2i} = w_{2i}^* - \gamma$), in which case the weight of the ensemble for feature $i$ is $w_{2i}^*$, a situation which occurs if $w_{1i}^*$ and $w_{2i}^*$ have the same sign. □

Table 1: Datasets, models and neighborhoods used in experiments. RF$\rightarrow$ Random Forest, NN$\rightarrow$ Neural Network and NB$\rightarrow$ Naive Bayes.

| Dataset | Modality | Black-box model acc/$R^2$, | Realistic neighborhood creation methods |
|---|---|---|---|
| IRIS | tabular | RF classifier, 93% | KDEGen (Botari et al., 2020), RF (Plumb et al., 2018) |
| MEPS | tabular | RF regressor, 0.325 | (Plumb et al., 2018) |
| FMNIST | image | NN classifier, 87% | VAEGen (Botari et al., 2020) |
| Rotten Tomatoes | text | NB classifier, 75% | Word2VecGen (Botari et al., 2020) |

**Implications of the Theorem 1:** The following are the main takeaways from Theorem 1: (1) If the signs of the explanations for unconstrained least squares for the two environments differ for some feature, then the algorithm outputs a zero as the attribution for that feature. (2) If the signs of the explanations for the two environments are the same, then the algorithm outputs the lesser magnitude of the two. These two properties are highly desirable from an algorithmic recourse or actionability perspective, where the first biases us to not rely on features where the black-box function changes direction rapidly (unidirectionality). The second, provides a reserved estimate so that we do not incorrectly over rely on the particular feature (stability).

**Behavior for More than Two Environments:** Given Assumptions 1 and 2 we now discuss the behavior of our method for more than two environments. If the number of environments is odd, then using similar logic to that discussed in the proof sketch one can see that the feature attribution would be equal to the median of the feature attributions across all the environments. Essentially, all environments with optimal least squares attributions above the median would be at $+\gamma$, while those below it would be at $-\gamma$. The one at the median would remain so with no incentive for any environment to alter its attribution making it a NE. This is a stable choice that is also likely to be faithful as we have no more information to decide otherwise. On the other hand if we have an even number of environments the situation is a bit different. The final attribution in this case depends on the middle two environments in the same manner as the two environment case proved in Theorem 1. Thus, if the optimal least squares attributions of the middle two environments have opposite sign, then the final attribution is zero, else its the lower of the two attributions in terms of the numerical value. This happens because the NE for the other environments is $+\gamma$ or $-\gamma$ depending on if their optimal least squares attributions are above or below those of the middle two environments. This again is a stable and likely to be faithful choice, where also unidirectionality is preferred.

## 5 EXPERIMENTS

We test our method on four real world datasets covering all three modalities: IRIS (Tabular) (Dheeru & Karra Taniskidou, 2017), Medical Expenditure Panel Survey (Tabular) (Agency for Healthcare Research and Quality, 2019), Fashion MNIST (Image) (Xiao et al., 2017), and Rotten Tomatoes reviews (Text) (Pang et al., 2002) with LIME-like random (rand) and MeLIME-like realistic neighborhood generation (real) or MAPLE-like realistic neighborhood selection (mpl). The summary of black-box classifier accuracies, and type of realistic perturbation used for the datasets are provided in Table 1. In all cases except FMNIST which comes with its own test partition we randomly split the datasets into 80/20% train/test partition and average results for the local explanations over this test partition. We chose FMNIST over other higher resolution image datasets as MeLIME performs best w.r.t. it, making it an acid test for our method. For LINEX we produce two environments where the two environments are formed by performing bootstrap sampling on the base environment which is created either by rand, real or mpl type neighborhood generation described above. Thus in all cases the union of the environments is the same as a single neighborhood used to produce explanations for the competitors making it a fair comparison. Behavior of LINEX with more environments is in Appendix C.

Given the neighborhood generation schemes we compare LINEX with LIME, Smoothed LIME (S-LIME), MeLIME and MAPLE, where for S-LIME we average the explanations of LIME across the LINEX environments. SHAP's results are in Appendix F, since it is not a natural fit here.

**Metrics:** We evaluate the methods using five simple metrics: Infidelity (INFD), Generalized Infidelity (GI), Coefficient Inconsistency (CI), Class Attribution Consistency (CAC) and Unidirectionality ($\Upsilon$). The first two evaluate *faithfulness*, the next two *stability* and the last *goodness for recourse*.

Figure 2: Sample results using FMNIST dataset for two classes. (a-c): Class *Dress*, (d-f): Class *Sandal*. (a, d): MeLIME explanations. (b, d): LINEX explanations. (c, f): Original images. We observe that LINEX explanations capture important artifacts and thus exhibit significantly higher correlation with the original images, where in aggregate too the correlations are high w.r.t. images belonging to a particular class, thus showcasing higher stability (i.e. high CAC) as is seen in Table 3. More examples are shown in Appendix E

Table 2: Below we see three example positive sentiment sentences from the Rotten Tomatoes dataset. Green and red indicate the most important word highlighted by MeLIME and LINEX respectively. As can be seen LINEX highlights stronger positive sentiment words. More examples in Appendix D.

| Example 1 | Example 2 | Example 3 |
|---|---|---|
| one-of-a-kind near-masterpiece | moving tale of love and destruction in unexpected places , unexamined lives | spare yet audacious . . . |

Let $D_t$ denote a (test) dataset with examples $(x, y)$ where $y_b(x)$ is the black-box models prediction on $x$ and $y_e^{x'}(x)$ is the prediction on $x$ ($\in \mathcal{X}$) using the explanation model at $x'$. The feature attributions (or coefficients) for the explanation model at $x$ are denoted by $c_e^x$, and $\mathcal{N}_x$ denotes the exemplar neighborhood of $x$ with $|.|_{\text{card}}$ denoting cardinality. We now define the above metrics.

*Infidelity (INFD):* This is the most commonly used metric to validate the faithfulness of explanation models (Ribeiro et al., 2016). Here we define it as the MAE between the black-box and explanation model predictions across all the test points: $\text{INFD} = \frac{1}{|D_t|_{\text{card}}} \sum_{(x,y) \in D_t} |y_b(x) - y_e^x(x)|$.

*Generalized Infidelity (GI):* This metric has also been used in previous works (Ramamurthy et al., 2020) to measure the generalizability of local explanations to neighboring test points. It is defined as: $\text{GI} = \frac{1}{|D_t|_{\text{card}}} \sum_{(x,y) \in D_t} \frac{1}{|\mathcal{N}_x|_{\text{card}}} \sum_{x' \in \mathcal{N}_x} |y_b(x) - y_e^{x'}(x)|$.

*Coefficient Inconsistency (CI):* This notion has been used before (Hancox-Li, 2020) to measure how robust an explanation method is. We define it to be the MAE between the the linear coefficients of all the test points and their respective neighbors: $\text{CI} = \frac{1}{|D_t|_{\text{card}}} \sum_{(x,y) \in D_t} \frac{1}{|\mathcal{N}_x|_{\text{card}}} \sum_{x' \in \mathcal{N}_x} |c_e^x - c_e^{x'}|_1$.

*Class-Attribution Consistency (CAC):* For local explanations of classification black-boxes, we expect certain important features to be highlighted across most of the explanations of a class. This is codified by this metric which is defined as follows: $\text{CAC} = \frac{1}{|\mathcal{Y}|_{\text{card}}} \sum_{y \in \mathcal{Y}} r(\mu_e^y, \mu_y)$, where $\mathcal{Y}$ denotes the set of class labels in the dataset, $\mu_y$ the mean (vector) of all inputs in class $y \in \mathcal{Y}$, $\mu_e^y$ the mean explanation for class $y$ and $r$ the Pearson's correlation coefficient. This metric quantifies the consistency between the important features for a class and attributions provided by the explanations.

*Coefficient Unidirectionality ($\Upsilon$):* This is motivated and defined in equation 2 in section 4.1.

We report the above metrics in Table 3. Each result in Table 3 is mean $\pm$ standard error of the mean over five kernel sizes $\tau\sqrt{d}$ generally, where $\tau = \{0.05, 0.1, 0.25, 0.5, 0.75\}$. Test neighborhoods do not make sense for random perturbations with FMNIST and Rotten Tomatoes because the features (viz. superpixels) used by neighboring test examples are different. Also, we do not use realistic perturbations with MEPS since KDE and VAE generators do not work well with categorical data. In addition, since MEPS data uses regression black-box, CAC cannot be computed. All these justify the missing entries in Table 3. The results were generated on Linux machines with 56 cores and 242 GB RAM. More details regarding the exact perturbation schemes for LIME/MeLIME/MAPLE, the perturbation neighborhood sizes and the time taken by the different methods are Appendix A and B.

**Observations:** We see that in terms of CAC, LINEX is better than baselines in all cases which indicates that on average the local explanations with LINEX highlight the important features characterizing the entire class making them more stable. This is further verified by looking at the $\Upsilon$ and CI

Table 3: Comparing the different methods using metrics infidelity (INFD), generalized infidelity (GI), coefficient inconsistency (CI), class attribution consistency (CAC) and unidirectionality ($\Upsilon$). $\uparrow$ indicates higher value for the metric is better, and $\downarrow$ indicates lower is better. Results better by $\geq 1\%$ are bolded. LINEX is better than baselines in 18 out of 35 cases, and worse only in 5 cases. Plots showing behavior of these metrics with varying neighborhood size, number of environments and kernel width are in Appendix C.

| Dataset | Method | INFD $\downarrow$ | GI $\downarrow$ | CI $\downarrow$ | $\Upsilon \uparrow$ | CAC $\uparrow$ |
|---|---|---|---|---|---|---|
| IRIS | LIME | $0.015 \pm 0.011$ | $0.132 \pm 0.042$ | $0.319 \pm 0.132$ | $0.646 \pm 0.040$ | $0.667 \pm 0.167$ |
| | S-LIME | $0.015 \pm 0.010$ | $0.077 \pm 0.011$ | $0.143 \pm 0.045$ | $0.704 \pm 0.037$ | $0.878 \pm 0.034$ |
| | LINEX/rand | $0.013 \pm 0.009$ | $\mathbf{0.052 \pm 0.008}$ | $\mathbf{0.044 \pm 0.013}$ | $\mathbf{0.802 \pm 0.043}$ | $\mathbf{0.921 \pm 0.042}$ |
| | MeLIME | $0.008 \pm 0.003$ | $0.049 \pm 0.018$ | $0.219 \pm 0.108$ | $0.629 \pm 0.013$ | $0.464 \pm 0.100$ |
| | LINEX/real | $0.009 \pm 0.003$ | $\mathbf{0.029 \pm 0.003}$ | $\mathbf{0.024 \pm 0.002}$ | $\mathbf{0.744 \pm 0.044}$ | $\mathbf{0.942 \pm 0.023}$ |
| | MAPLE | $0.009 \pm 0.001$ | $0.038 \pm 0.004$ | $0.261 \pm 0.033$ | $0.458 \pm 0.032$ | $0.586 \pm 0.035$ |
| | LINEX/mpl | $0.013 \pm 0.000$ | $\mathbf{0.020 \pm 0.000}$ | $\mathbf{0.026 \pm 0.002}$ | $\mathbf{0.694 \pm 0.008}$ | $\mathbf{0.929 \pm 0.004}$ |
| MEPS | LIME | $0.158 \pm 0.066$ | $0.214 \pm 0.041$ | $0.005 \pm 0.001$ | $0.981 \pm 0.006$ | NA |
| | S-LIME | $0.158 \pm 0.066$ | $0.214 \pm 0.042$ | $0.005 \pm 0.001$ | $0.974 \pm 0.008$ | |
| | LINEX/rand | $\mathbf{0.130 \pm 0.052}$ | $\mathbf{0.164 \pm 0.021}$ | $0.003 \pm 0.001$ | $0.979 \pm 0.006$ | |
| | MAPLE | $\mathbf{0.063 \pm 0.000}$ | $\mathbf{0.067 \pm 0.000}$ | $0.007 \pm 0.000$ | $0.957 \pm 0.000$ | NA |
| | LINEX/mpl | $0.098 \pm 0.001$ | $0.094 \pm 0.001$ | $0.007 \pm 0.000$ | $0.950 \pm 0.000$ | |
| FMNIST | LIME | $0.162 \pm 0.003$ | NA | NA | NA | NA |
| | S-LIME | $0.142 \pm 0.003$ | | | | |
| | LINEX/rand | $0.149 \pm 0.002$ | | | | |
| | MeLIME | $\mathbf{0.001 \pm 0.000}$ | $\mathbf{0.277 \pm 0.000}$ | $0.007 \pm 0.000$ | $0.769 \pm 0.000$ | $0.327 \pm 0.000$ |
| | LINEX/real | $0.100 \pm 0.002$ | $0.304 \pm 0.001$ | $0.002 \pm 0.000$ | $\mathbf{0.780 \pm 0.000}$ | $\mathbf{0.649 \pm 0.001}$ |
| Rotten Tomatoes | LIME | $0.079 \pm 0.036$ | NA | NA | NA | NA |
| | S-LIME | $0.075 \pm 0.035$ | | | | |
| | LINEX/rand | $0.069 \pm 0.032$ | | | | |
| | MeLIME | $\mathbf{0.029 \pm 0.001}$ | $0.391 \pm 0.000$ | $0.000 \pm 0.000$ | $0.999 \pm 0.000$ | $0.909 \pm 0.000$ |
| | LINEX/real | $0.053 \pm 0.000$ | $\mathbf{0.361 \pm 0.000}$ | $0.000 \pm 0.000$ | $1.000 \pm 0.000$ | $\mathbf{0.953 \pm 0.001}$ |

metrics where LINEX is similar or better than others. For GI and INFD metrics, the results are more evenly spread which implies that LINEX's main advantage is obtaining stable and unidirectional explanations that are faithful to a similar degree.

We show an example MeLIME and LINEX/realistic explanation on FMNIST in Figure 2, where we see that LINEX explanations are more coherent and also delineate the outline of the image more clearly compared to MeLIME explanations leading to better CAC. Even on the text data we see more reasonable attributions in Table 2, where "masterpiece", "moving" and "audacious" are highlighted as the most important words indicative of positive sentiment in the three examples.

An interesting observation is that when it comes to the stability metrics (CI and CAC) and unidirectionality LINEX with even random perturbation model is better than MeLIME in some cases. This is very promising as it means LINEX could be potentially be trusted without the need to generate realistic perturbations which may be computationally expensive or not even possible.

## 6  DISCUSSION

In this paper we have provided a method based on a game theoretic formulation and inspired by the invariant risk minimization principle to provide faithful, stable and unidirectional explanations. We have defined the latter property and argued that it is somewhat of a necessity (may not be sufficient) for recourse. We have theoretically shown that our method has a strong tendency to be stable and unidirectional as we will mostly eliminate features where the black-box models gradient changes abruptly and in other cases choose a conservative value. Empirically, we have verified this where we outperform competitors in majority of the cases on these metrics. An interesting observation is also that in some cases our method provides more stable and unidirectional explanations with just a random perturbation model relative to more expensive methods that use realistic neighbors.

In the future, it would be worth experimenting with more varied strategies to form environments and if possible find the optimal ones (Creager et al., 2020), which may lead to picking even more relevant features that are "causal" to the local decision.

ETHICS STATEMENT

With the wide adoption of deep learning technologies, explaining or understanding the reasons behind their decisions has become extremely important in many critical applications (Arya et al., 2019). Numerous explainability methods have been proposed in literature to explain individual decisions of black-box models (Ribeiro et al., 2016; Plumb et al., 2018; Botari et al., 2020; Dhurandhar et al., 2018a; Lundberg & Lee, 2017). Although LINEX is more stable and undirectional than other competing approaches, it still is a posthoc explainability method that may not be completely faithful to the black-box model. This of course is not just a limitation of our approach, but nonetheless should be taken into account before a user makes a decision. Our method could also be used to divulge information by exposing the inner workings of the black-box leading to privacy concerns. One possible mitigation strategy in this case would be to keep the sensitive attributes hidden from the explainer.

REPRODUCIBILITY STATEMENT

Experimental details are provided in Section 5 of the main paper and Appendix B. All datasets are public. Code will be provided during the discussion phase through an anonymized link.

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

APPENDIX

## A  EFFICIENCY OF LINEX

It is important to note that the query complexity (i.e. number of times we query the black box to obtain an explanation) of LINEX is the same as that of LIME since the union of the environments is the same as a LIME perturbation neighborhood. This is important in todays cloud-driven world where models may exist on different cloud platforms and posthoc explanations are an independent service where each call to the model has an associated cost. In terms of running time for two environments, convergence was fast and running time was approximately 2.5 times that of LIME (LINEX took 2.5 seconds on IRIS for 30 examples as opposed to 1 second by LIME, LINEX took 47 seconds on MEPS for 500 examples as opposed to 18 seconds by LIME), which is very similar to Smoothed LIME (S-LIME) (took 2.3 seconds on IRIS and 40 seconds on MEPS) that we still outperform in majority of the cases.

Realistic neighborhood generation can be time consuming especially for MeLIME since generators have to be trained which may take up to an hour using a single GPU for datasets such as FMNIST. After the generator is trained and neighborhood sampled MeLIME takes the same amount of time as LIME since the model fitting procedure is the same. MAPLE took 1.5 seconds for the IRIS dataset for 30 examples and 27 seconds for 500 MEPS examples.

## B  EXPERIMENTAL DETAILS

### B.1  DATASET DETAILS AND HYPERPARAMETER SPECIFICATIONS

We describe the datasets and the hyperparameters used for each. We set perturbation neighborhood sizes 10 (IRIS), 500 (MEPS), 100 (FMNIST-random), 500 (FMNIST-realistic), 100 (Rotten tomatoes) for generating local explanations. We also use 3, 10, 10, 5 as examplar neighborhood sizes to compute GI, CI and $\Upsilon$ metrics for the four datasets respectively. We also use $5-$sparse explanations for all cases except FMNIST with realistic perturbations where we follow MeLIME and generate a dense explanation using ridge penalty with penalty multiplier value of $0.001$. The $\ell_\infty$ bound $\gamma$ in Algorithm 1 is set as the maximum absolute value of linear coefficient computed by running LIME/MeLIME in the two individual environments. Please look at IRIS dataset first since it contains some of the common details used across others.

**IRIS (Tabular):**  This dataset has 150 instances with four numerical features representing the sepal and petal width and length in centimeters. The task is to classify instances of Iris flowers into three species: *setosa*, *versicolor*, and *virginica*. A random forest classifier was trained with a train/test split of 0.8/0.2 and yielded a test accuracy of 93%. We provide local explanations for the prediction probabilities for class *setosa*. For both random and realistic perturbations, we use a perturbation neighborhood size of $n$. For random perturbations, we used the same approach followed by LIME and sample from a Gaussian around each data point. Realistic perturbations (with the same number $n$) were generated using KDEGen Botari et al. (2020), a kernel density estimator (KDE) with the Gaussian kernel fitted on the training dataset to sample data around a sample point. For both random and realistic perturbations, we weight the neighborhood using a Gaussian kernel of width $\tau\sqrt{d}$, where $d$ is the dimension of the feature vector and $\tau = \{0.05, 0.1, 0.25, 0.5, 0.75\}$, and this corresponded to kernel widths $\{0.1, 0.2, 0.5, 1.0, 1.5\}$. We also perform a weighted version of realistic selection where we use MAPLE Plumb et al. (2018) to assign weights to all the test examples and pick the top $n$ weighted examples to use as the perturbation neighborhood. For random/realistic perturbations and realistic selection, the corresponding environments (of size $n$ each) for LINEX are created by drawing $k$ bootstrap samples where $k = \{2, 3, 4, 5\}$ in our experiments. We test for $n = \{10, 20, 30, 40, 50\}$ with this dataset.

**Medical Expenditure Panel Survey (Tabular):**  The Medical Expenditure Panel Survey (MEPS) dataset is produced by the US Department of Health and Human Services. It is a collection of surveys of families of individuals, medical providers, and employers across the country. We choose *Panel 19* of the survey which consists of a cohort that started in 2014 and consisted of data collected over 5 rounds of interviews over $2014-2015$. The outcome variable was a composite utilization feature that

quantified the total number of healthcare visits of a patient. The features used included demographic features, perceived health status, various diagnosis, limitations, and socioeconomic factors. We filter out records that had a utilization (outcome) of 0, and log-transformed the outcome for modeling. These pre-processing steps resulted in a dataset with 11136 examples and 32 categorical features. We train a random forest regressor that has a test $R^2$ of $0.325$ in this dataset. We provide local explanations of the predictions. With MEPS, we do not use realistic perturbations since KDE and VAE generators do not work well with categorical data. Otherwise the setting is similar as IRIS data, except that we use $n = \{50, 100, 200, 300, 400, 500\}$. The kernel widths in this case were $\{0.28, 0.57, 1.41, 2.83, 4.24\}$. We use $k = \{2, 3, 4, 5\}$ for this dataset.

**Fashion MNIST (Images):** This dataset has $28 \times 28$ grayscale images of fashion articles with 60,000 train and 10,000 test samples. The task is to classify these into 10 classes corresponding to coat, shoe, and so on. A neural network trained with test accuracy of 87%. Explanations are generated for the prediction probabilities corresponding to the predicted class for each example. We choose 1000 test examples to generate explanations. Realistic perturbations were generated using VAEGen (Botari et al., 2020), a Variational Auto Encoder (VAE) fitted on the training dataset. For random perturbations, the we chose $n$ from $\{50, 100, 200, 300, 400, 500\}$ and kernel sizes were $\{0.43, 0.85, 2.14, 4.27, 6.41\}$. For realistic perturbations we chose $n$ from $\{250, 500, 750, 1000\}$ and the kernel widths were $\{1.4, 2.8, 7.0, 14.0, 21.0\}$. We use $k = \{2, 3, 4, 5\}$ for this dataset.

**Rotten Tomatoes (Text):** This dataset contains 10662 movie reviews from rotten tomatoes website along with their sentiment polarity, i.e., positive or negative reviews and the task is to classify the sentiment of the reviews into positive or negative. The review sentences were vectorized using CountVectorizer and TfidfTransformer and a sklearn Naive Bayes classifier was fitted on training dataset which yielded a test accuracy of 75%. Explanations are generated for the prediction probabilities corresponding to the predicted class for each example. Realistic perturbations were generated using Word2VecGen (Botari et al., 2020), wherein word2vec embeddings are first trained using the training corpus and new sentences are generated by randomly replacing a sentence word whose distance in the embedding space lies within the radius of the neighbourhood. For both random and realistic perturbations, $n$ was chosen from $\{25, 50, 75, 100\}$. The kernel sizes were $\{0.42, 1.06, 2.12, 3.18\}$ for random perturbations (kernel size $0.21$ resulted in numerical issues), and $\{0.21, 0.42, 1.06, 2.12, 3.18\}$ for realistic perturbations. We use $k = \{2, 3, 4, 5\}$ for this dataset.

## C  Results with All Datasets and Hyperparameter Combinations for Random and Realistic Perturbations

We present results with all hyperparameter combinations for random and realistic perturbations. Results for LIME with random perturbations (LIME), smoothed LIME (S-LIME), LINEX with random perturbations (LINEX/rand), MeLIME (MeLIME), LINEX with MeLIME-like realistic neighborhoods (LINEX/real), MAPLE (MAPLE), LINEX with MAPLE-like realistic neighborhoods (LINEX/mpl) are presented in figures 4-18. The legend for these figures are given in Figure 3.

For the four datasets, we perform ablations by varying one of perturbation neighborhood size (Figures 4-8), number of environments (Figures 9-13), and kernel width (Figures 14-18). Each point in these figures are averaged over all possible values for the two parameters that are not ablated. For example, each point in Figure 4 is averaged over all possible values for kernel widths and number of environments for a given perturbation neighborhood size. Standard errors of the mean are also plotted in the same color with lesser opacity. Lower values of Infidelity (INFD), Generalized Infidelity (GI), Coefficient Inconsistency (CI) are better whereas for Unidirectionality ($\Upsilon$) and Class Attribution Consistency (CAC) higher values are better.

Figures 4-8 show ablations with respect to perturbation neighborhood sizes. Considering all datasets, the stability/recourse metrics (CI, $\Upsilon$, CAC) are clearly better for LINEX compared to its counterparts. For LINEX methods (LINEX/rand, LINEX/real, LINEX/mpl), the metrics get better or stays approximately the same generally as perturbation neighborhood size increases keeping with the intuition that larger perturbation neighborhood sizes should produce explanations that are more stable in the exemplar neighborhood. $\Upsilon$ for FMNIST is somewhat of an exception which could be because of the high dimensionality as well as the quality of MeLIME perturbations.

Turning to the fidelity metrics (INFD and GI) in tabular datasets, we see that the results still favor LINEX, but less heavily compared to the stability/recourse metrics. This is in line with what we observe in Table 3. In IRIS and MEPS, LINEX is close to or outperforms the corresponding baselines in the GI measure (except for LINEX/mpl with MEPS). This gap closes a bit with INFD, but we note that GI is a better measure since it estimates how faithful explanations are in a exemplar neighborhood. With the text dataset, LINEX variants are slightly more favored, whereas with the image dataset, the baselines have an edge.

Considering Figures 9-13, we see that variations are less stark with respect to number of environments overall for LINEX variants. Note that except for S-LIME, other baselines do not use multiple environments, and hence stay constant. The slight variations in MAPLE are due to the effect of random seeds. In the stability/recourse metrics, again LINEX variants emerge as the clear winner across datssets. With the faithfulness metrics (GI and INFD), in the text dataset, LINEX variants generally perform better, whereas the baselines have a better performance in the image dataset.

Finally, we study the variation of the performance measures with respect to kernel width in Figures 14-18. We see that the stability/recourse metrics flatten out in all cases with large kernel widths. This behaviour holds true for faithfulness metrics (GI and INFD) as well except in some cases. GI and INFD measures also increase before they flatten out since the fit becomes poorer at larger kernel widths. The stability/recourse metrics become better or remain approximately the same since explanations generally improve or preserve their stability properties as kernel widths increase. Note that very small kernel widths can lead to unexpected behavior that does not fit the trend as seen with the tabular datasets since explanations can become hyper-local. MAPLE and LINEX/mpl stay the same at different kernel widths since they use a different weighting scheme. As with other ablations, we see that LINEX variants are similar or better in stability/recourse metrics overall, while with the faithfulness metrics the results are more mixed.

Note that we do not compute MeLIME perturbations with MEPS since KDE and VAE generators do not work well with categorical data, and do not use compute CAC since the task is regression. Further, the features used in explanations for different test examples are not comparable for random perturbations with FMNIST and Rotten Tomatoes, hence we cannot compute CAC for those cases as well. This explains the missing curves/plots.

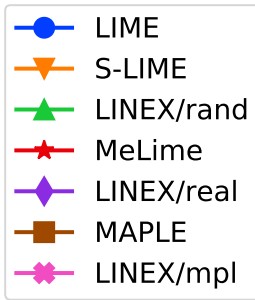

Figure 3: Legend for figures 4-18

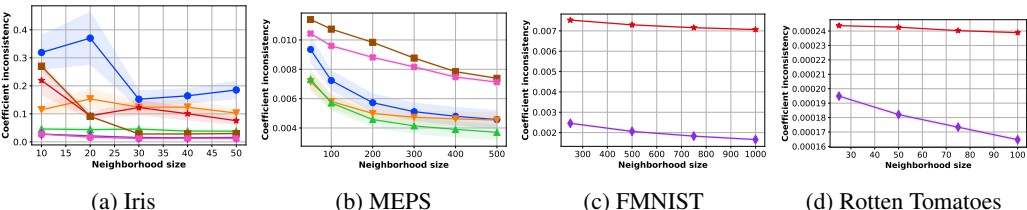

| (a) Iris | (b) MEPS | (c) FMNIST | (d) Rotten Tomatoes |

Figure 4: Coefficient inconsistency (CI) vs. Perturbation neighborhood size.

(a) Iris     (b) FMNIST     (c) Rotten Tomatoes

Figure 5: Class attribution consistency (CAC) vs. Perturbation neighborhood size.

(a) Iris    (b) MEPS    (c) FMNIST    (d) Rotten Tomatoes

Figure 6: Unidirectionality ($\Upsilon$) vs. Perturbation neighborhood size.

(a) Iris    (b) MEPS    (c) FMNIST    (d) Rotten Tomatoes

Figure 7: Generalized infidelity (GI) vs. Perturbation neighborhood size.

(a) Iris    (b) MEPS    (c) FMNIST (random)    (d) FMNIST (realistic)

(e) Rotten Tomatoes (random)     (f) Rotten Tomatoes (realistic)

Figure 8: Infidelity (INFD) vs. Perturbation neighborhood size.

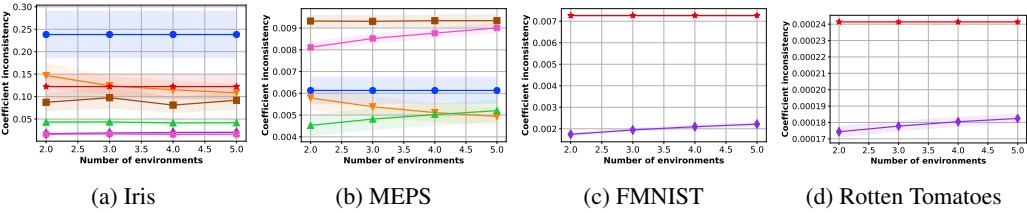

(a) Iris    (b) MEPS    (c) FMNIST    (d) Rotten Tomatoes

Figure 9: Coefficient inconsistency (CI) vs. Number of environments.

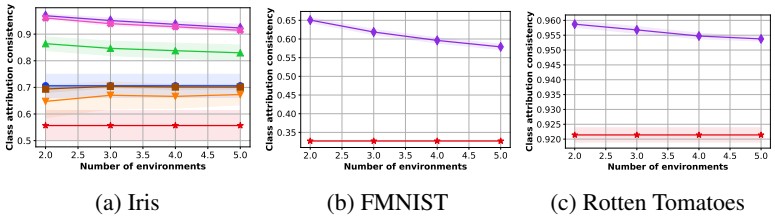

Figure 10: Class attribution consistency (CAC) vs. Number of environments.

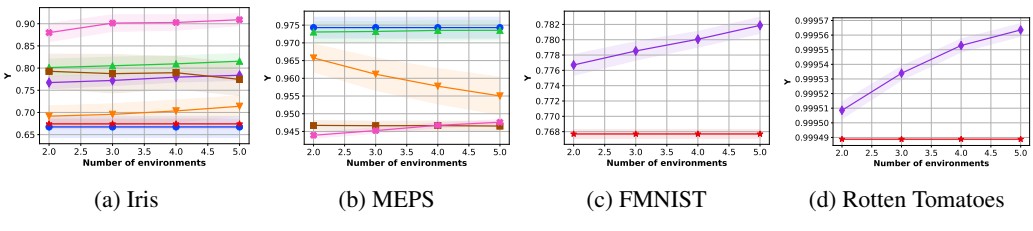

Figure 11: Unidirectionality ($\Upsilon$) vs. Number of environments.

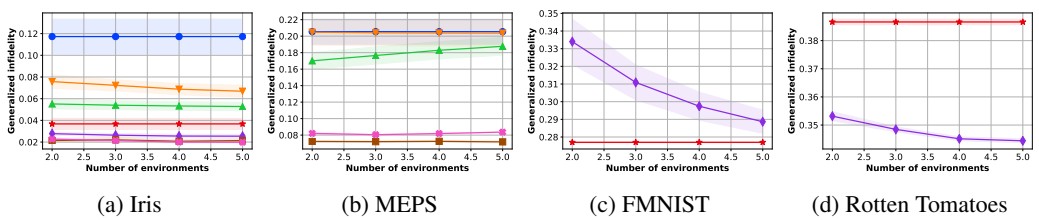

Figure 12: Generalized infidelity (GI) vs. Number of environments.

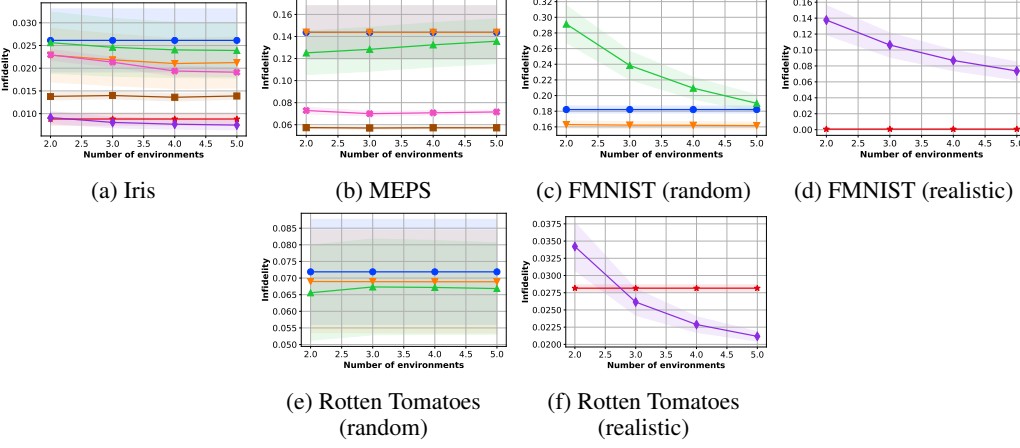

Figure 13: Infidelity (INFD) vs. Number of environments.

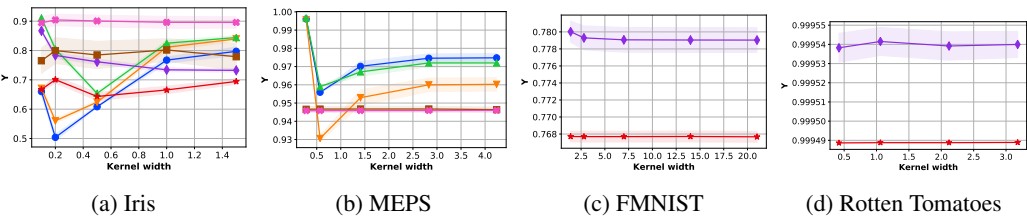

Figure 14: Unidirectionality ($\Upsilon$) vs. Kernel width.

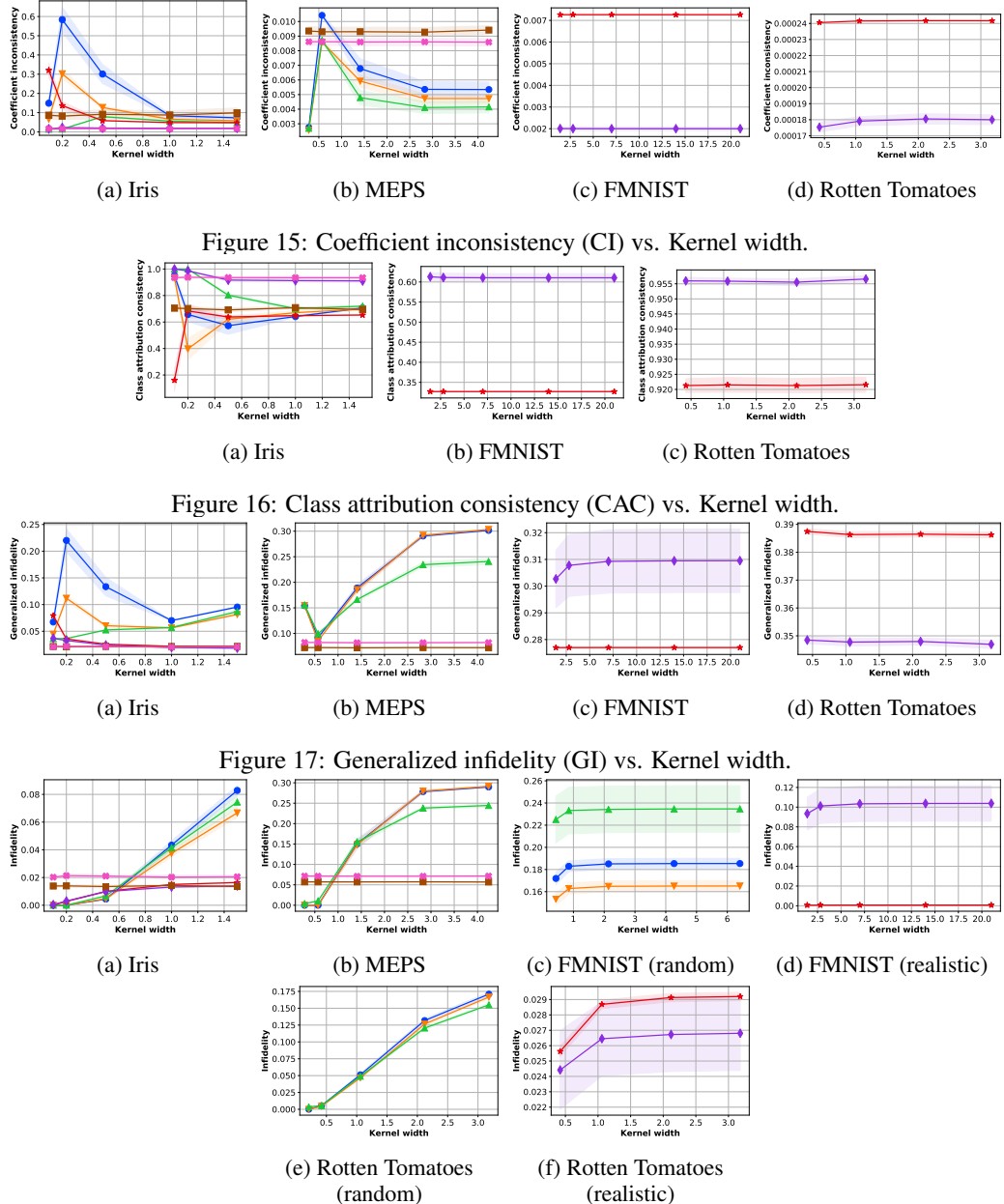

Figure 15: Coefficient inconsistency (CI) vs. Kernel width.

Figure 16: Class attribution consistency (CAC) vs. Kernel width.

Figure 17: Generalized infidelity (GI) vs. Kernel width.

Figure 18: Infidelity (INFD) vs. Kernel width.

## D EXAMPLE FEATURE ATTRIBUTIONS IN TEXT DATA: MELIME VS LINEX

Below we see sample attributions by the two methods along with the magnitude of the attributions. Attribution magnitudes are printed with a precision of $10^{-3}$ and shown along with the corresponding words in descending order.

### D.1 POSITIVE SENTIMENT

```
enticing and often funny documentary .
MeLIME: documentary funny and enticing often
LINEX : documentary funny often enticing and
MeLIME: 0.517 0.446 0.333 0.317 0.311
```

```
LINEX : 0.416 0.377 0.342 0.331 0.330

one-of-a-kind near-masterpiece .
MeLIME: kind near masterpiece
LINEX : masterpiece kind one
MeLIME: 0.832 0.695 0.182
LINEX : 0.712 0.384 0.381

a fast , funny , highly enjoyable movie .
MeLIME: enjoyable highly funny fast movie
LINEX : enjoyable highly fast funny movie
MeLIME: 0.550 0.432 0.412 0.389 0.198
LINEX : 0.409 0.389 0.372 0.350 0.326

ferrara's strongest and most touching movie of recent years .
MeLIME: touching years most strongest and
LINEX : touching most recent strongest and
MeLIME: 0.735 0.490 0.450 0.443 0.427
LINEX : 0.490 0.488 0.450 0.444 0.407

saved from being merely way-cool by a basic , credible compassion .
MeLIME: cool basic credible merely from
LINEX: cool credible merely compassion from
MeLIME: 1.514 0.050 0.040 0.029 0.026
LINEX : 0.358 0.308 0.304 0.299 0.293

really quite funny .
MeLIME: funny quite really
LINEX : funny quite really
MeLIME: 0.559 0.417 0.233
LINEX : 0.462 0.368 0.275

spare yet audacious . . .
MeLIME: spare yet audacious
LINEX : audacious spare yet
MeLIME: 0.626 0.447 0.395
LINEX : 0.501 0.431 0.422

an engrossing and infectiously enthusiastic documentary .
MeLIME: engrossing documentary and enthusiastic an
LINEX : engrossing documentary an enthusiastic and
MeLIME: 0.593 0.455 0.358 0.354 0.333
LINEX : 0.461 0.407 0.374 0.357 0.350

a wildly funny prison caper .
MeLIME: funny caper wildly prison
LINEX : funny caper prison wildly
MeLIME: 0.541 0.364 0.214 0.193
LINEX : 0.403 0.335 0.245 0.239

this charming but slight tale has warmth , wit
and interesting characters compassionately portrayed .
MeLIME: charming compassionately and interesting portrayed
LINEX : charming compassionately has tale portrayed
MeLIME: 0.690 0.507 0.456 0.444 0.424
LINEX : 0.464 0.435 0.431 0.430 0.429

thoughtful , provocative and entertaining .
MeLIME: thoughtful entertaining and provocative
```

```
LINEX : thoughtful entertaining and provocative
MeLIME: 0.612 0.517 0.402 0.395
LINEX : 0.505 0.461 0.415 0.404

the film is quiet , threatening and unforgettable .
MeLIME: quiet unforgettable and film the
LINEX : unforgettable quiet film and is
MeLIME: 0.597 0.483 0.412 0.325 0.303
LINEX : 0.421 0.416 0.388 0.378 0.338

a moving tale of love and destruction in unexpected places , unexamined lives .
MeLIME: unexpected moving love tale lives
LINEX : moving unexpected places lives in
MeLIME: 0.692 0.662 0.577 0.538 0.499
LINEX : 0.538 0.530 0.521 0.513 0.501

though frodo's quest remains unfulfilled , a hardy group of
determined new zealanders has proved its creative mettle .
MeLIME: creative group proved has new
LINEX : creative quest its proved determined
MeLIME: 0.602 0.441 0.424 0.402 0.393
LINEX : 0.410 0.392 0.390 0.385 0.381
```

## D.2 NEGATIVE SENTIMENT

```
originality is sorely lacking .
MeLIME: lacking sorely is originality
LINEX : lacking sorely originality is
MeLIME: 0.543 0.381 0.296 0.278
LINEX : 0.430 0.356 0.314 0.271

an ugly , pointless , stupid movie .
MeLIME: stupid pointless ugly movie an
LINEX : stupid pointless ugly movie an
MeLIME: 0.543 0.499 0.385 0.365 0.276
LINEX : 0.446 0.411 0.373 0.360 0.350

so devoid of pleasure or sensuality that it cannot even be dubbed hedonistic .
MeLIME: devoid even be dubbed of
LINEX : devoid so dubbed be cannot
MeLIME: 0.666 0.416 0.413 0.372 0.344
LINEX : 0.400 0.392 0.387 0.380 0.368

neither revelatory nor truly edgy--merely crassly flamboyant
and comedically labored .
MeLIME: edgy neither nor labored revelatory
LINEX : edgy neither nor labored truly
MeLIME: 1.256 0.338 0.277 0.204 0.021
LINEX : 0.439 0.398 0.398 0.369 0.349

occasionally funny , sometimes inspiring , often boring .
MeLIME: boring occasionally inspiring sometimes often
LINEX : boring occasionally sometimes often inspiring
MeLIME: 0.669 0.242 0.218 0.210 0.182
LINEX : 0.377 0.266 0.266 0.250 0.236

a cumbersome and cliche-ridden movie greased
with every emotional device known to man .
MeLIME: cliche every device movie with
```

```
LINEX : cliche every man cumbersome emotional
MeLIME: 0.695 0.449 0.327 0.280 0.268
LINEX : 0.385 0.361 0.354 0.349 0.309

ponderous , plodding soap opera disguised as a feature film .
MeLIME: plodding soap ponderous opera disguised
LINEX : plodding soap film ponderous feature
MeLIME: 0.579 0.522 0.421 0.408 0.382
LINEX : 0.442 0.440 0.418 0.406 0.377

kitschy , flashy , overlong soap opera .
MeLIME: soap flashy opera overlong kitschy
LINEX : soap flashy opera overlong kitschy
MeLIME: 0.499 0.397 0.391 0.358 0.230
LINEX : 0.389 0.362 0.360 0.346 0.300

[a] poorly executed comedy .
MeLIME: poorly comedy executed
LINEX : poorly comedy executed
MeLIME: 0.653 0.348 0.257
LINEX : 0.502 0.335 0.309

a bad movie that happened to good actors .
MeLIME: bad happened movie to that
LINEX : bad happened to movie actors
MeLIME: 0.692 0.396 0.371 0.367 0.242
LINEX : 0.442 0.384 0.367 0.361 0.344

a complete waste of time .
MeLIME: waste complete time of
LINEX : waste complete time of
MeLIME: 0.614 0.425 0.313 0.247
LINEX : 0.480 0.381 0.348 0.278

don't waste your money .
MeLIME: waste money don your
LINEX : waste money don your
MeLIME: 0.592 0.497 0.408 0.309
LINEX : 0.483 0.450 0.411 0.337

witless and utterly pointless .
MeLIME: pointless witless and utterly
LINEX : pointless witless utterly and
MeLIME: 0.652 0.491 0.263 0.245
LINEX : 0.506 0.444 0.311 0.269
```

## E    EXAMPLE FEATURE ATTRIBUTIONS IN IMAGE DATA: MeLIME VS LINEX

We show feature attributions for individual example images with MeLIME and LINEX with MeLIME perturbations in Figure 19. In Figure 20 we show class-wise mean feature attributions along with mean images. Clearly, LINEX explanations provide more meaningful feature attributions.

## F    RESULTS FOR ALL METHODS INCLUDING SHAP

In Table 4, we provide the results for SHAP along with all methods for easy comparison. Note that SHAP does not have standard errors since it is computed only once per test point. The INFD values for SHAP are miniscule since SHAP values add up to the predictions by definition. In order to

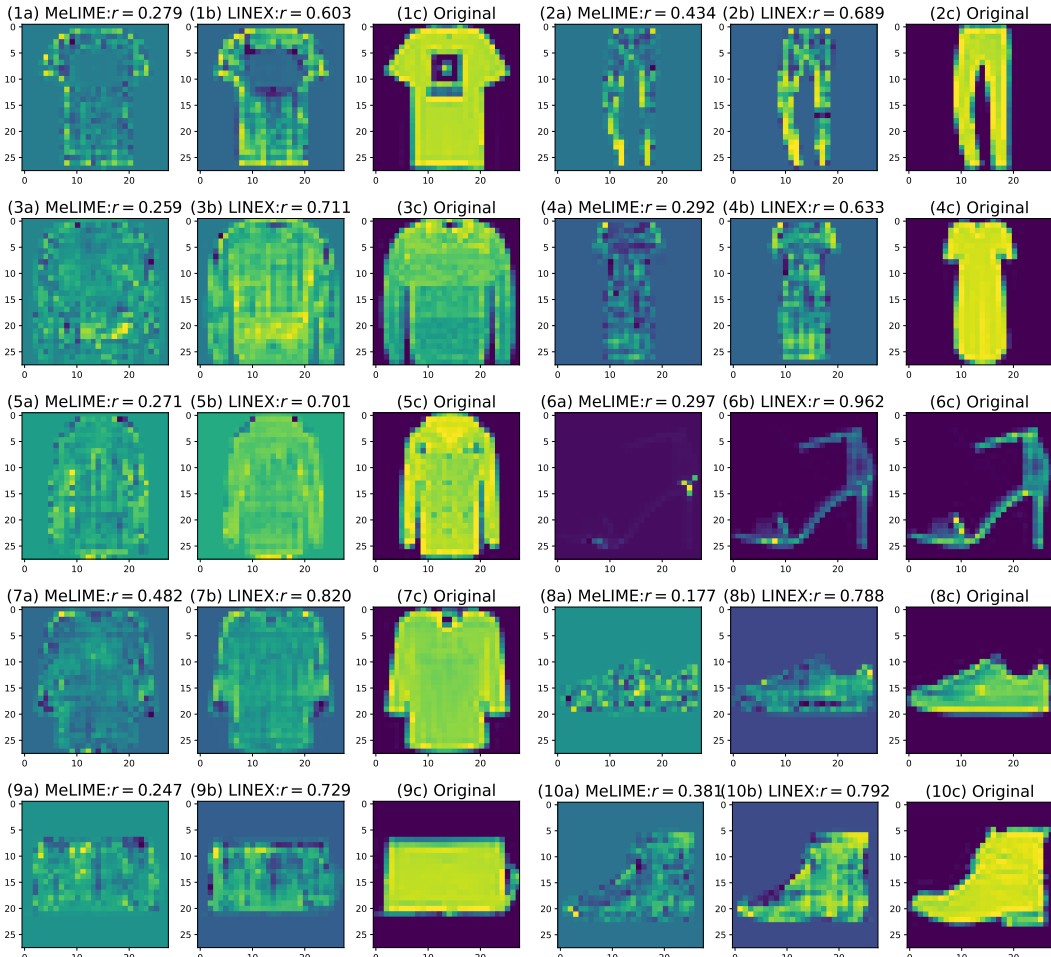

Figure 19: Results using individual samples for realistic perturbations for FMNIST dataset for all classes:1-10 (*T-shirt/top, Trouser, Pullover, Dress, Coat, Sandal, Shirt, Sneaker, Bag* and *Ankle boot*). (a) MeLIME feature attributions for an image. (b) LINEX feature attributions for an image. (c) Original image in the class. The $r$ values show Pearson's correlation between feature attributions and the original image from the respective class. We observe that LINEX attributions/explanations exhibit significantly higher correlation with the original image belonging to a particular class (i.e. high CAC).

compute GI, CI, $\Upsilon$, CAC, we convert the SHAP values to SHAP attributions (Amparore et al., 2021, equation 4) first and follow the same approach used by other explanation methods.

# G  ERROR ANALYSIS OF LINEX

We perform error analysis for LINEX to gain better understanding about the method. We choose FMNIST dataset for doing this since: (a) This is the highest dimensional dataset (784 dimensions) that we have. (b) LINEX/real under performs MeLIME in terms of the INFD measure here (see Table 5) more heavily compared to other datasets. (c) It is easier to visualize the explanations for this dataset.

We start by observing that even though LINEX/real underperforms in the INFD metric, the gap is not so great in the GI metric, which suggests that MeLIME may be overfitting explanations here. We also note that in terms of CI, $\Upsilon$, and CAC metrics, LINEX/real clearly outperforms MeLIME.

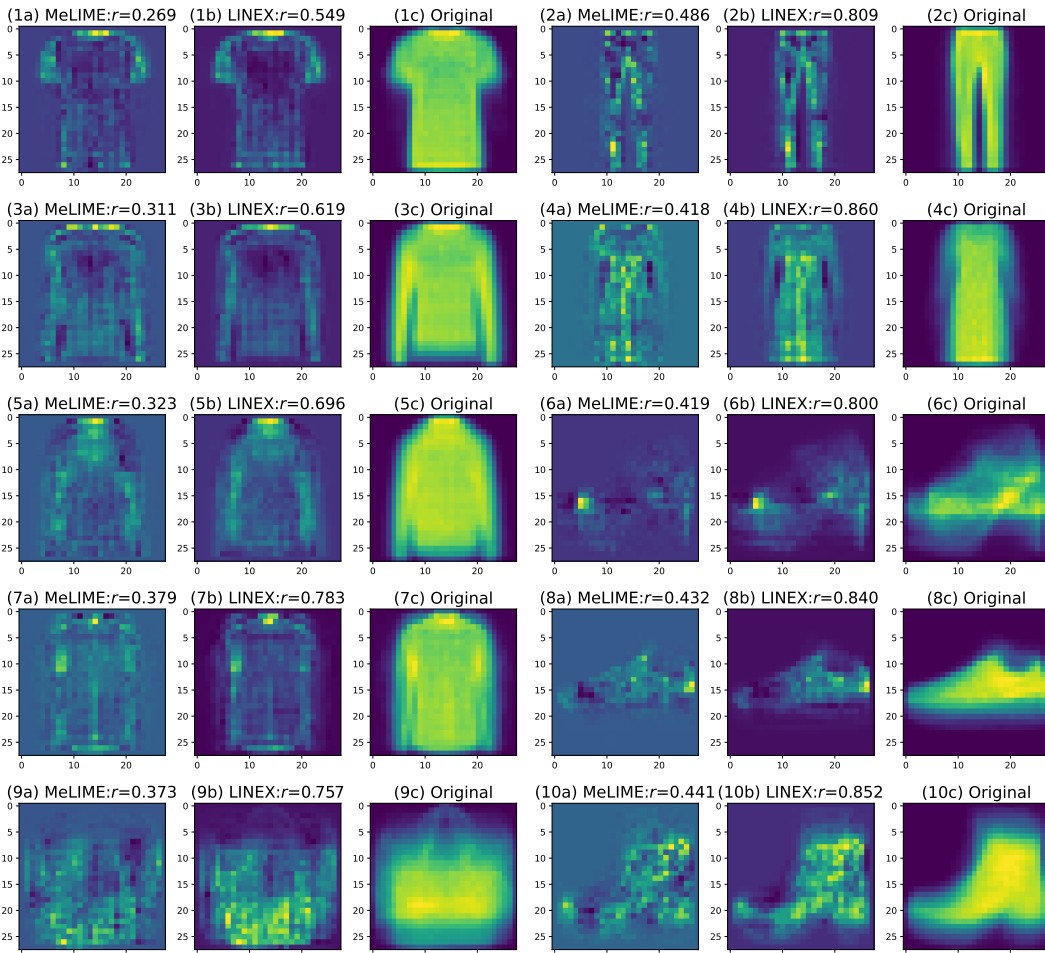

Figure 20: Results using realistic perturbations for FMNIST dataset with mean feature importances for all classes:1-10 (*T-shirt/top, Trouser, Pullover, Dress, Coat, Sandal, Shirt, Sneaker, Bag* and *Ankle boot*). (a) Mean feature attributions of all images in the class using MeLIME. (b) Mean feature attributions of all images in the class using LINEX. (c) Mean of all images in the class. The $r$ values show Pearson's correlation between average feature attributions and mean of the original images from the respective classes. We observe that LINEX explanations/attributions exhibit significantly higher correlation with the original images belonging to a particular class (i.e. high CAC).

We now choose a sample of images from the dataset where LINEX/real has highest instance-level infidelity numbers and display them in Figure 21. Just looking at the explanations and the corresponding original images visually, it is evident that LINEX/real highlights the prominent features like stripes in the t-shirt, handles of the bags, outlines of the boots/shoes, even though the infidelity values are high. However, MeLIME misses out on some of these prominent features and focuses only on optimizing the fit. The fact that LINEX zeroes in on important features also provides additional evidence for the closeness of GI metrics between the two methods, and the better performance of LINEX/real with CI, $\Upsilon$, and CAC metrics.

Table 4: Comparing the different methods using metrics infidelity (INFD), generalized infidelity (GI), coefficient inconsistency (CI), class attribution consistency (CAC) and unidirectionality ($\Upsilon$). $\uparrow$ indicates higher value for the metric is better, and $\downarrow$ indicates lower is better. Results better by $\geq$ 1% are bolded. LINEX is better than baselines in 18 out of 35 cases, and worse only in 5 cases. We set neighborhood sizes 10 (IRIS), 500 (MEPS), 100 (FMNIST-random), 500 (FMNIST-realistic), 100 (Rotten tomatoes) for generating local explanations. We also use 3, 10, 10, 5 as exemplar neighborhood sizes to compute GI, CI and $\Upsilon$ metrics for the four datasets respectively.

| Dataset | Method | INFD $\downarrow$ | GI $\downarrow$ | CI $\downarrow$ | $\Upsilon$ $\uparrow$ | CAC $\uparrow$ |
|---|---|---|---|---|---|---|
| IRIS | LIME | $0.015 \pm 0.011$ | $0.132 \pm 0.042$ | $0.319 \pm 0.132$ | $0.646 \pm 0.040$ | $0.667 \pm 0.167$ |
| | S-LIME | $0.015 \pm 0.010$ | $0.077 \pm 0.011$ | $0.143 \pm 0.045$ | $0.704 \pm 0.037$ | $0.878 \pm 0.034$ |
| | LINEX/rand | $0.013 \pm 0.009$ | $\mathbf{0.052 \pm 0.008}$ | $\mathbf{0.044 \pm 0.013}$ | $\mathbf{0.802 \pm 0.043}$ | $\mathbf{0.921 \pm 0.042}$ |
| | MeLIME | $0.008 \pm 0.003$ | $0.049 \pm 0.018$ | $0.219 \pm 0.108$ | $0.629 \pm 0.013$ | $0.464 \pm 0.100$ |
| | LINEX/real | $0.009 \pm 0.003$ | $\mathbf{0.029 \pm 0.003}$ | $\mathbf{0.024 \pm 0.002}$ | $\mathbf{0.744 \pm 0.044}$ | $\mathbf{0.942 \pm 0.023}$ |
| | MAPLE | $0.009 \pm 0.001$ | $0.038 \pm 0.004$ | $0.261 \pm 0.033$ | $0.458 \pm 0.032$ | $0.586 \pm 0.035$ |
| | LINEX/mpl | $0.013 \pm 0.000$ | $\mathbf{0.020 \pm 0.000}$ | $\mathbf{0.026 \pm 0.002}$ | $\mathbf{0.694 \pm 0.008}$ | $\mathbf{0.929 \pm 0.004}$ |
| | SHAP | 0.007 | 0.197 | 0.248 | 0.664 | 0.524 |
| MEPS | LIME | $0.158 \pm 0.066$ | $0.214 \pm 0.041$ | $0.005 \pm 0.001$ | $0.981 \pm 0.006$ | |
| | S-LIME | $0.158 \pm 0.066$ | $0.214 \pm 0.042$ | $0.005 \pm 0.001$ | $0.974 \pm 0.008$ | NA |
| | LINEX/rand | $\mathbf{0.130 \pm 0.052}$ | $\mathbf{0.164 \pm 0.021}$ | $0.003 \pm 0.001$ | $0.979 \pm 0.006$ | |
| | MAPLE | $\mathbf{0.063 \pm 0.000}$ | $\mathbf{0.067 \pm 0.000}$ | $0.007 \pm 0.000$ | $0.957 \pm 0.000$ | NA |
| | LINEX/mpl | $0.098 \pm 0.001$ | $0.094 \pm 0.001$ | $0.007 \pm 0.000$ | $0.950 \pm 0.000$ | |
| | SHAP | 0.000 | 0.091 | 0.009 | 0.940 | NA |
| FMNIST | LIME | $0.162 \pm 0.003$ | | | | |
| | S-LIME | $0.142 \pm 0.003$ | NA | NA | NA | NA |
| | LINEX/rand | $0.149 \pm 0.002$ | | | | |
| | MeLIME | $\mathbf{0.001 \pm 0.000}$ | $\mathbf{0.277 \pm 0.000}$ | $0.007 \pm 0.000$ | $0.769 \pm 0.000$ | $0.327 \pm 0.000$ |
| | LINEX/real | $0.100 \pm 0.002$ | $0.304 \pm 0.001$ | $0.002 \pm 0.000$ | $\mathbf{0.780 \pm 0.000}$ | $\mathbf{0.649 \pm 0.001}$ |
| | SHAP | 0.000 | 1.962 | 0.589 | 0.551 | 0.038 |
| Rotten Tomatoes | LIME | $0.079 \pm 0.036$ | | | | |
| | S-LIME | $0.075 \pm 0.035$ | NA | NA | NA | NA |
| | LINEX/rand | $0.069 \pm 0.032$ | | | | |
| | MeLIME | $\mathbf{0.029 \pm 0.001}$ | $0.391 \pm 0.000$ | $0.000 \pm 0.000$ | $0.999 \pm 0.000$ | $0.909 \pm 0.000$ |
| | LINEX/real | $0.053 \pm 0.000$ | $\mathbf{0.361 \pm 0.000}$ | $0.000 \pm 0.000$ | $1.000 \pm 0.000$ | $\mathbf{0.953 \pm 0.001}$ |
| | SHAP | 0.000 | 0.384 | 0.008 | 0.999 | 0.015 |

Table 5: Comparing the different methods using metrics infidelity (INFD), generalized infidelity (GI), coefficient inconsistency (CI), class attribution consistency (CAC) and unidirectionality ($\Upsilon$). $\uparrow$ indicates higher value for the metric is better, and $\downarrow$ indicates lower is better. Results that are statistically significantly better based on paired t-test are bolded. LINEX is better than baselines in 19 out of 35 cases, and worse only in 6 cases. Plots showing behavior of these metrics with varying neighborhood size, number of environments and kernel width are in Appendix C.

| Dataset | Method | INFD $\downarrow$ | GI $\downarrow$ | CI $\downarrow$ | $\Upsilon$ $\uparrow$ | CAC $\uparrow$ |
|---|---|---|---|---|---|---|
| IRIS | LIME | $0.015 \pm 0.011$ | $0.132 \pm 0.042$ | $0.319 \pm 0.132$ | $0.646 \pm 0.040$ | $0.667 \pm 0.167$ |
| | S-LIME | $0.015 \pm 0.010$ | $0.077 \pm 0.011$ | $0.143 \pm 0.045$ | $0.704 \pm 0.037$ | $0.878 \pm 0.034$ |
| | LINEX/rand | $0.013 \pm 0.009$ | $\mathbf{0.052 \pm 0.008}$ | $\mathbf{0.044 \pm 0.013}$ | $\mathbf{0.802 \pm 0.043}$ | $\mathbf{0.921 \pm 0.042}$ |
| | MeLIME | $0.008 \pm 0.003$ | $0.049 \pm 0.018$ | $0.219 \pm 0.108$ | $0.629 \pm 0.013$ | $0.464 \pm 0.100$ |
| | LINEX/real | $0.009 \pm 0.003$ | $\mathbf{0.029 \pm 0.003}$ | $\mathbf{0.024 \pm 0.002}$ | $\mathbf{0.744 \pm 0.044}$ | $\mathbf{0.942 \pm 0.023}$ |
| | MAPLE | $\mathbf{0.009 \pm 0.001}$ | $0.038 \pm 0.004$ | $0.261 \pm 0.033$ | $0.458 \pm 0.032$ | $0.586 \pm 0.035$ |
| | LINEX/mpl | $0.013 \pm 0.000$ | $\mathbf{0.020 \pm 0.000}$ | $\mathbf{0.026 \pm 0.002}$ | $\mathbf{0.694 \pm 0.008}$ | $\mathbf{0.929 \pm 0.004}$ |
| MEPS | LIME | $0.158 \pm 0.066$ | $0.214 \pm 0.041$ | $0.005 \pm 0.001$ | $0.981 \pm 0.006$ | |
| | S-LIME | $0.158 \pm 0.066$ | $0.214 \pm 0.042$ | $0.005 \pm 0.001$ | $0.974 \pm 0.008$ | NA |
| | LINEX/rand | $0.130 \pm 0.052$ | $0.164 \pm 0.021$ | $\mathbf{0.003 \pm 0.001}$ | $0.979 \pm 0.006$ | |
| | MAPLE | $\mathbf{0.063 \pm 0.000}$ | $\mathbf{0.067 \pm 0.000}$ | $0.007 \pm 0.000$ | $0.957 \pm 0.000$ | NA |
| | LINEX/mpl | $0.098 \pm 0.001$ | $0.094 \pm 0.001$ | $0.007 \pm 0.000$ | $0.950 \pm 0.000$ | |
| FMNIST | LIME | $0.162 \pm 0.003$ | | | | |
| | S-LIME | $0.142 \pm 0.003$ | NA | NA | NA | NA |
| | LINEX/rand | $0.149 \pm 0.002$ | | | | |
| | MeLIME | $\mathbf{0.001 \pm 0.000}$ | $\mathbf{0.277 \pm 0.000}$ | $0.007 \pm 0.000$ | $0.769 \pm 0.000$ | $0.327 \pm 0.000$ |
| | LINEX/real | $0.100 \pm 0.002$ | $0.304 \pm 0.001$ | $\mathbf{0.002 \pm 0.000}$ | $\mathbf{0.780 \pm 0.000}$ | $\mathbf{0.649 \pm 0.001}$ |
| Rotten Tomatoes | LIME | $0.079 \pm 0.036$ | | | | |
| | S-LIME | $0.075 \pm 0.035$ | NA | NA | NA | NA |
| | LINEX/rand | $0.069 \pm 0.032$ | | | | |
| | MeLIME | $\mathbf{0.029 \pm 0.001}$ | $0.391 \pm 0.000$ | $0.000 \pm 0.000$ | $0.999 \pm 0.000$ | $0.909 \pm 0.000$ |
| | LINEX/real | $0.053 \pm 0.000$ | $\mathbf{0.361 \pm 0.000}$ | $0.000 \pm 0.000$ | $\mathbf{1.000 \pm 0.000}$ | $\mathbf{0.953 \pm 0.001}$ |

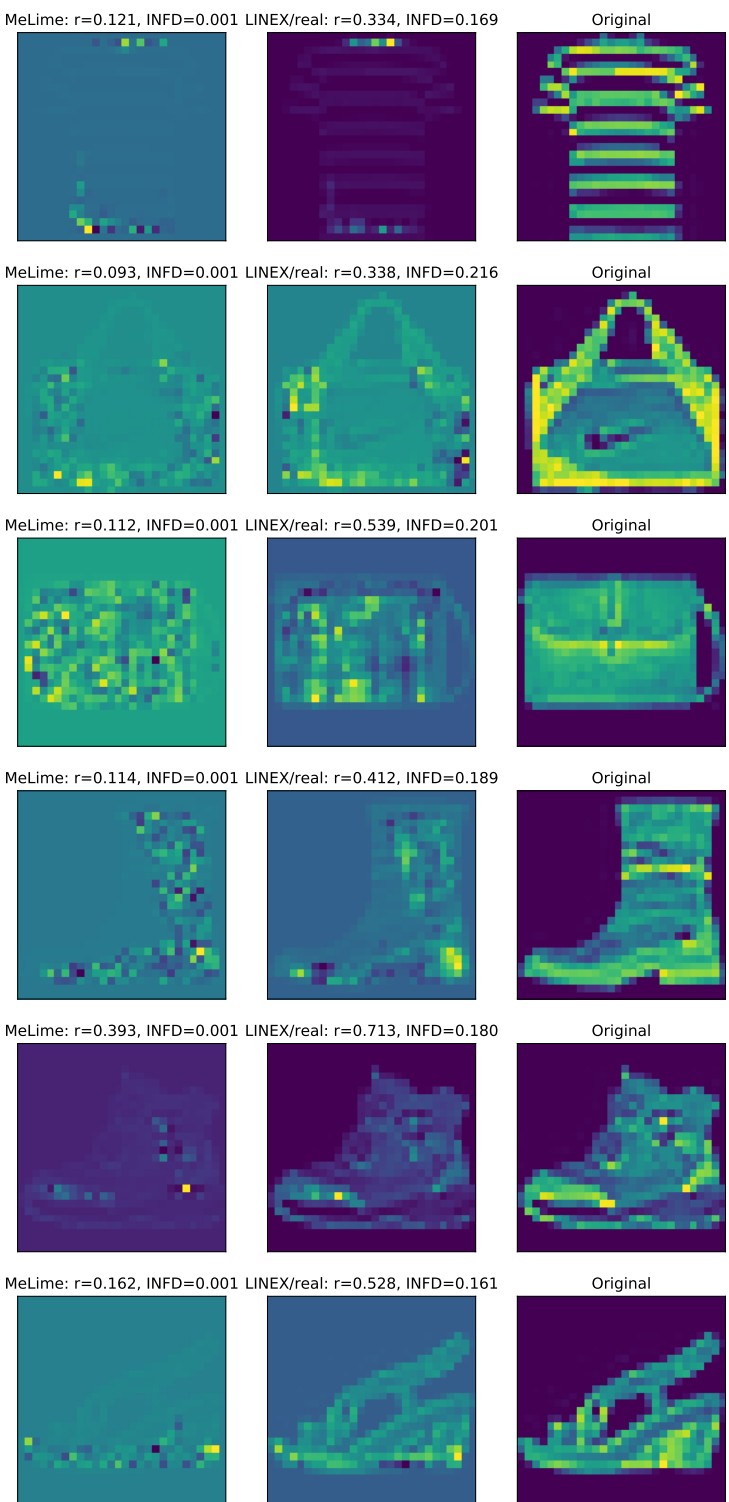

Figure 21: Error analysis for a chosen set of examples in FMNIST using MeLIME and LINEX/real methods. The three columns are the MeLIME feature attributions, LINEX/real feature attributions, and the original images. The rows correspond to different examples. We show the Pearson's correlation coefficient between feature attributions and mean of the original images from the respective classes ($r$) and instance-level infidelity (INFD) measures. LINEX seems to highlight important features like stripes in the t-shirt, handles of the bags, outlines of the boots/shoes more prominently, while MeLIME seems to overfit to the data while missing out on highlighting some key features prominently.

