# OpenReview forum: "Locally Invariant Explanations: Towards Causal Explanations through Local Invariant Learning"
_ICLR.cc/2022/Conference — ICLR 2022 Submitted_

### Official Review · Reviewer_9WKN · 2021-11-03

**Correctness:** 3
**Technical Novelty And Significance:** 2
**Empirical Novelty And Significance:** 2
**Recommendation:** 5
**Confidence:** 4

**Main Review:**

One concern about creation of the local environments. Faithfulness of the local predictor to the original model has commonly be a concern in local explanation methods. Creating “environments” around a data unit further exacerbates this concern. How can one assure the faithfulness of the predictors in each environment to the original model? Even if one can empirically show that such approach works, without a theoretical guarantee, one cannot rely on LINEX. On the other hand, one other related concern is the answer to the following question: Hoe exactly to create environments? The authors themselves acknowledge that this is important and propose two ways of doing so: Random perturbation, and realistic generation according to the literature that they cite. IMO, one major concern is solving this issue. Pointing to ongoing work about this problem does not suffice. This paper is not clearly stating whether they are proposing a solid solution on how to create local environments.

The title of the paper suggests that this paper provides causal explanations. The main text attempts at slightly maneuvering over the terminology of causality. The abstract claims that the work ascertains explanatory features without using a causal graph. All of these evidence point toward using causal inference and/or explaining the black box model with causal features. However, the paper falls short of providing such explanations. In fact, to my understanding, the authors do not extract causal features. It would be great if the authors clarify on when/where they extract causal features, or alternatively, modify their writing not to reflect causal inference.

 The constraint in the optimization of the proposed algorithm does not necessarily formulate the situation of a Nash equilibrium. I am not clear about the theoretical properties of the Nash equilibrium in this problem. If one wants to interpret the results using concepts of this equilibrium, the whole problem needs to be formally formulated and axiomatically translated.

 The authors have shown their results on more than one environment (<= 5) in the Appendix. I am wondering about the behavior on even more environments? How would the authors recommend choosing the number of environments? I have an understanding that this is, at the moment, determined only empirically, and am not clear if there is any theoretical analysis over the number of environments.

On the application side, the paper is showing interesting results.


**Summary Of The Paper:**

This paper approaches the problem of locally explaining predictions of black box models. They attempt to do so by creating local environments around a data unit. They propose their method, LINEX to overcome the shortcomings of the method, LIME. LINEX learns locally invariant predictors and aggregates their explanation for locally explaining the original model. The authors show some theoretical properties as well as empirical evidence.

**Summary Of The Review:**

The paper lacks in novelty, contribution, and supporting theory of the proposed methodology.

---

> ### Author Response · Authors · 2021-11-16
> **Response to 9WKN**
>
> 1) **LINEX and faithfulness:** It is important to note that as with any posthoc method (viz. LIME, MeLIME, etc.) ours too minimizes a loss w.r.t. the black-box models outputs hence trying to be faithful to the original model. Also note that the overall model which is outputted by our algorithm needs to be faithful, unidirectional and stable w.r.t. the black-box and not necessarily each of the environment specific models as those are just components to produce the final model. The latter two properties are more important as one can produce many models that have the same fidelity or are equally faithful. Hence, in Theorem 1, we prove under certain conditions, that in fact our model will produce unidirectional and stable explanations. This hasn't been shown in prior explainability works. As practical situations might violate some of these assumptions we also test our method on real datasets covering all three modalities where notably, we are never worse than the competitors w.r.t. these metrics (i.e. unidirectionality and stability ones).
> We hope the reviewer appreciates this fact.
>
> 2) **Environment creation:** We have provided and tested on largely two options to create environments: i) Random and ii) Realistic. The first environment in both cases is simply the neighborhood produced by standard methods such as LIME/MeLIME/MAPLE. Every successive environment is produced using bootstrap sampling (mentioned in section 5). Hence, we have a standard process to create successive environments once the neighborhood (or first environment) is created. Of course, having a realistic neighborhood is preferable as is also evidenced in the experiments, but as mentioned in the paper, it may not always be possible to do so due to limited data or computational constraints. In such cases one could create a random neighborhood as it is easier to do so. However, as seen in the experiments in some cases (and mentioned in section 5 last paragraph), even with random neighborhood generation LINEX does better than MeLIME which uses realistic generation with regards to unidirectionality and stability, which is notable. In summary, we would suggest the following procedure for environment generation: i) Create a realistic neighborhood, ii) if (i) is not possible, create a random neighborhood, iii) consider this neighborhood as the first environment following which perform bootstrap sampling $k$ times to create $k$ additional environments each of equal size. Furthermore, improved solutions to generate environments could come out of future research, which we do not believe undermines our current proposal as we already outperform the competitors on many of these metrics.
>
> 3) **Why causal:** Causality (Pearl, 2009) typically tries to identify factors that are part of a global data generating process. In the situation of local explainability we care about factors that the black-box model utilizes to make a local decision. Hence, all globally causal factors may not be relevant to a locality. For example, consider a black-box model $f$ that bases its decisions on two features $x_1, x_2$
> in one locality and only on $x_1$ in the other. More precisely, let $f=g_1(x_1,x_2)$ and $f=g_2(x_1)$ be the models in each locality respectively, where $g_1$ and $g_2$ are the local data generators/functions. Globally, both features $x_1$ and $x_2$ are causal to the model. However, from a local standpoint in the second locality only $x_1$ is causal. It is possible because of random neighborhood generation and other instabilities that LIME type of methods might incorrectly pick up both $x_1$ and $x_2$ as the relevant features in this locality. However, given the conservative behavior of our approach described in section 4.3 our method is more likely to pick just $x_1$ which presumably has a more persistent correlation with the output of the model. This stability is witnessed in the experiments. This is also the reason why we titled our paper "...Towards Causal..." as we are more likely to uncover the true local factors in a local data generating DAG, nonetheless were careful to refrain from saying point blank "...Causal explanations..." as that we agree would be a much bolder claim. As such, if you still believe the title is somewhat misleading we would be open to replacing the word "Causal" with say "Robust" or "Robust and Intuitive" or "Stable and Unidirectional" or some other suggestion you might have.

---

> > ### Author Response · Authors · 2021-11-16
> > **Response to reviewer 9WKN continued...**
> >
> > 4) **Why NE?**  The setup of our problem mimics a game where each environment is a player, the utility is the (negative) mean squared loss and the strategy is each $\tilde{w}_i$ learned for each environment. Given that the utility is convex as well as the norm constraints, this implies that the game is a concave game, for which it is well known that pure NEs will always exist (Debreu, 1952). Our Theorem 1 provides such a NE for this game under certain conditions. In other words, our setup in algorithm 1 which is a simultaneous constrained optimization is a well defined game. The solution to the game is a pure (generalized) NE (https://www.genconv.org/files/Kaohsiung_Aussel2.pdf). The addition of the constraints makes the action sets of the players coupled making it a generalized NE.
> >
> > 5) **Choosing number of environments:** Typically, we would plot the behavior of our method w.r.t. metrics we care about as done in the appendix and choose the correct number of environments. In many case, we see that results are stable with just two environments. Theoretically, given our bootstrap sampling procedure to create environments one could compute the (max) amount of redundancy or overlap that one might find on average for a new bootstrap sample over any of the ones already seen. If the overlap is likely to be large then there is little value in having the new sample a.k.a. environment. For instance, even with two environments the overlap is $> 63$\% on average. Hence, with more environments this number should increase rapidly implying that few environments should be sufficient in most cases.

---

### Official Review · Reviewer_388N · 2021-11-04

**Correctness:** 4
**Technical Novelty And Significance:** 2
**Empirical Novelty And Significance:** 2
**Recommendation:** 5
**Confidence:** 4

**Main Review:**

Strengths

+ LINEX is a robust form of LIME, without losing much fidelity
+ The motivation for choosing the IRM framework is laid out well
+ The experiments are over a wide range of datasets and evaluation metrics.

Concerns:
- LINEX doesn’t seem to be significantly better than MeLime or MAPLE, especially for non-tabular data. MAPLE is competitive on the MEPS dataset, while MeLime shows competitive performance on the FMNIST and Rotten Tomatoes datasets. When considering that LINEX takes 2.5x the training time as LIME (as mentioned in Appendix A), did other methods have the same benefit of additional training time/iterations?
- It is not clear why the paper states ‘Causal’ in the title (and also elsewhere in the paper), this seems unnecessary. The paper uses the word ‘causal’ loosely, and there is no support to the claim that LINEX explanations are causal.
- The paper states that ‘unidirectionality’ is a new property. However, there have been papers in recent years on robust explanations and attributional robustness (see [3],[4],[5],[6] references below). How is this different from these efforts?
- The novelty of the work seems limited, as the algorithm and theoretical results in the paper are largely based on [1],[2].

[1] Ahuja, Kartik et al. “Invariant Risk Minimization Games.” ArXiv abs/2002.04692, 2020
[2] Ahuja, Kartik et al. “Linear Regression Games: Convergence Guarantees to Approximate Out-of-Distribution Solutions.” ArXiv abs/2010.15234, 2021
[3] Lakkaraju, Himabindu et al. “Robust and Stable Black Box Explanations.” ArXiv abs/2011.06169, 2020
[4] Zhao, Xingyu et al. “BayLIME: Bayesian Local Interpretable Model-Agnostic Explanations.” ArXiv abs/2012.03058, 2020
[5] Chen, et al, Robust Attributional Regularization,  NeurIPS 2019
[6] Sarkar, Anindya et al, Enhanced Regularizers for Attributional Robustness, AAAI’21


**Summary Of The Paper:**

The paper proposes an algorithm LINEX for learning a robust LIME-like explanation for a black box model. The algorithm is based on the Invariant Risk Minimization Games framework proposed by [1],[2]. LIME is notorious for being very sensitive to its hyperparameters, and the paper posits that a robust variant of LIME is one which is invariant to these hyperparameters. The primary contribution of the paper is to formulate the perturbation-neighbourhoods of the point to be explained as ‘environments’ in an IRM setup. The paper also notes that characteristics of such invariant predictors (Theorem 2 of [2]) make them well-suited for explanations.

**Summary Of The Review:**

Utilizing invariances for robust explanations is an interesting direction, but the paper relies heavily on the algorithms and results presented in [1],[2]. The results, though competitive, are not striking enough to warrant the usage of LINEX over existing methods (and also over other formulations for robust explanations such as [3],[4] in the above references).

---

> ### Author Response · Authors · 2021-11-16
> **Response to reviewer 388N**
>
> 1) **Performance w.r.t. MeLIME and MAPLE:** It is important to note that LINEX is either better or similar (never worse) to both these baselines in terms of stability and unidirectionality metrics in all the experiments, which is our main motivation for proposing it. Over all metrics and datasets LINEX is better than MeLIME in 8 cases and worse only in 3 cases, while it is better than MAPLE in 4 cases and worse only in 2. Qualitatively this is maintained in Table 5 in the appendix where we now report statistically significant results (LINEX is again never worse than MeLIME or MAPLE on stability or unidirectionality metrics). We believe this is a worthwhile contribution especially given the novel perspective we bring to the subfield of local explanability.
>
> 2) **Training time for other methods:**
> Realistic neighborhood generation can be time consuming especially for MeLIME since generators have to be trained which may take up to an hour using a single GPU for datasets such as FMNIST. After the generator is trained and neighborhood sampled MeLIME takes the same amount of time as LIME since the model fitting procedure is the same. MAPLE took 1.5 seconds for the IRIS dataset for 30 examples and 27 seconds for 500 MEPS examples. We have now updated Section A (appendix) in the paper with these details.
>
> 3) **Title:** Causality (Pearl, 2009) typically tries to identify factors that are part of a global data generating process. In the situation of local explainability we care about factors that the black-box model utilizes to make a local decision. Hence, all globally causal factors may not be relevant to a locality. For example, consider a black-box model $f$ that bases its decisions on two features $x_1, x_2$
> in one locality and only on $x_1$ in the other. More precisely, let $f=g_1(x_1,x_2)$ and $f=g_2(x_1)$ be the models in each locality respectively, where $g_1$ and $g_2$ are the local data generators/functions. Globally, both features $x_1$ and $x_2$ are causal to the model. However, from a local standpoint in the second locality only $x_1$ is causal. It is possible because of random neighborhood generation and other instabilities that LIME type of methods might incorrectly pick up both $x_1$ and $x_2$ as the relevant features in this locality. However, given the conservative behavior of our approach described in section 4.3 our method is more likely to pick just $x_1$ which presumably has a more persistent correlation with the output of the model. This stability is witnessed in the experiments. This is also the reason why we titled our paper "...Towards Causal..." as we are more likely to uncover the true local factors in a local data generating DAG, nonetheless were careful to refrain from saying point blank "...Causal explanations..." as that we agree would be a much bolder claim. As such, if you still believe the title is somewhat misleading we would be open to replacing the word "Causal" with say "Robust" or "Robust and Intuitive" or "Stable and Unidirectional" or some other suggestion you might have.
>
> 4) **Novelty of Unidirectionality Metric:** Thanks for mentioning relevant works such as [3], [4], [5] and [6]. However, none of these works propose, evaluate or would directly ensure unidirectionality. [3] is a method directed towards stability using adversarial training which however is known to be unstable in practice (Ranganathan et al. 2019). It is also limited to tabular data with no publicly available code. [4] is mainly about a Bayesian version of LIME using MAP estimation and with an uninformative prior (which is all we would have in most cases) its behavior should be similar to S-LIME. [5]
>  and [6] try to learn, primarily in the image domain, attributions that are adversarially robust. Their goal is still complimentary to unidirectionality as one could have adversarially robust attributions for a specific input, yet these attributions could vary in sign (and magnitude) across near by inputs, thus leading to low unidirectionality.

---

> > ### Author Response · Authors · 2021-11-16
> > **Response to reviewer 388N continued...**
> >
> > 5) **Novelty w.r.t. [1], [2]:** It is important to realize that [1] and [2] are methods designed for the OOD generalization setting where the difference of this setting to ours is described in section 3. The main similarity of these works to ours is only that they also are game theory based approaches, but with the details being quite different. For one, they assume accessibility to environments and with assumptions on the structural causal model derive results on how the true causal factors could be divulged. In our case, we propose ways to generate environments as they are not given and have $l_1$ and $l_{\infty}$ constraints on the entire and environment specific parts of the model respectively, which is not the case with these prior works. As such algorithms in [1] and [2] do not try to produce unidirectional models that are also sparse and hence consumable. Moreover, the perspective we provide is novel in the context of local posthoc explanations where a priori it is not obvious that approaches from OOD generalization could be extended and adapted. We hope the reviewer appreciates this aspect as well.
> >
> > 6) **Contrast with [3], [4]:** As mentioned above, [3] is a method that uses adversarial training, which is known to be unstable in practice (Ranganathan et al. 2019). This situation is exacerbated for local explanations and something we found in our own experience. They also do not have code available publicly and we were unable to obtain it from the authors. Moreover, the method is primarily applicable to only tabular data. [4] is mainly about a Bayesian version of LIME using MAP estimation and with an uninformative prior (which is all we would have in most cases) its behavior should be similar to S-LIME (which we compare against) as bayesian counterparts of frequentist approaches typically result in averaging/smoothing (Mockus, 2012).

---

### Official Review · Reviewer_tbk2 · 2021-11-05

**Correctness:** 3
**Technical Novelty And Significance:** 3
**Empirical Novelty And Significance:** 3
**Recommendation:** 8
**Confidence:** 4

**Main Review:**

- I think this paper is well motivated and offers good contributions.
- Experimental results appear to strongly support the efficacy of the proposed method. The proposed method outperforms LIME, S-LIME and MeLime in over half the experiments, while achieving comparable performance on most that remain. However, results on FMNIST and Rotten Tomatoes appear to not be as convincing as the results on tabular data. The paper is also missing discussion on this aspect.
- I didn't see an error analysis of the proposed method, especially for cases where the performance is worse than the baselines. That's something that I think could add to the paper quite a bit.
- Overall the paper is well written and the presentation is quite clear. I do have some minor comments that I have listed below.

------------------------------------------------------------------------
Additional comments and questions:
- Page 3 local explainability setup vs IRM setup: "we want to highlight features in our explanation that may be spurious from the domain perspective, but nonetheless the black-box model uses them to make decisions". I think this is a wrong assertion that the underlying model "uses them to make decisions". These are post-hoc explanations and per my understanding, nowhere in the paper do you actually establish that the model indeed only relies on the explanations you've identified. Am I missing something?
- Section 4.3, Assumption 1: Is this independence across dimensions in the same environment or the same dimension across environments? Current phrasing doesn't disambiguate but I'm assuming it's the latter? Please rephrase.
- Section 4.3 Assumption 2: I'm not quite sure I follow when you say that Assumption 2 ensures "that we closely analyze the role of the l_\inf penalty. Please elaborate.

-------------------------------------------------------
Presentation and typos:
- I'm not quite sure the title ("Towards Causal Explanations") is appropriate. To the best of my understanding, I could not identify any causal estimand being estimated in this paper.
- Introduction Line 6: "based on a image scan" -> "based on an image scan"
- Introduction Page 2 first paragraph: "There have been variants suggested to overcome..." Please expand upon this sentence and give some additional background to the reader.
- Section 4.2.1: "optimized subject two constraints" -> "optimized subject to two constraints"
- Section 4.2.1: "In other words, for features where there is massive disagreement in even the direction of their impact are eliminated by our method" There is an agreement error here. Rephrase it as "In other words, features that have a massive disagreement in even the direction of their impact are eliminated by our method." Further, what does "massive" even mean here? Better to use grounded language or not use adjectives at all and just say disagreement.
- Missing related work: Jesse Vig, Sebastian Gehrmann, Yonatan Belinkov, Sharon Qian, Daniel Nevo, Simas Sakenis, Jason Huang, Yaron Singer, and Stuart Shieber. "Causal mediation analysis for interpreting neural nlp: The case of gender bias." NeurIPS 2020.

**Summary Of The Paper:**

Update:
In our back and forth, the authors have addressed or committed to address the concerns I had with this work. As a result, I have updated my score

---------------------------------------------

This paper introduces a new model agnostic local explanation method based on a game-theoretic formulation to ensure that resulting explanations are high fidelity and unidirectional across nearby examples. The idea is to first form several local environments via random perturbations or using creating realistic neighbors using generative or retrieval methods. Then, using the proposed algorithm, we iteratively learn a constrained least squares predictor for each environment and the final predictor is then the sum of these individual predictors. Experiments on three data modalities (tabular, vision, and text) show that the proposed method leads to better explanations than existing methods such as LIME.

**Summary Of The Review:**

Overall I think the paper provides a nice and sound contribution. The results are convincing on the tabular data but I would like some discussion about other modalities where results are somewhat comparable across several metrics. In addition, I have a few points I'd like some clarity on.

---

> ### Author Response · Authors · 2021-11-16
> **Response to reviewer tbk2**
>
> 1) **Error analysis where LINEX under performs:** We have now performed error analysis for LINEX using the FMNIST dataset since, (a) this is the highest dimensional dataset (784 dimensions) that we have, (b) LINEX/real under performs MeLIME in terms of the INFD measure here (see Table 5) more heavily compared to other datasets and (c) it is easier to visualize the explanations for this dataset. This analysis is reported in Section G (appendix) of the updated paper, where after looking at the worst 100 INFD examples for LINEX we visualize six among the top examples. Using these examples in Figure 21, we explain how LINEX seems to highlight qualitatively important features more prominently even for these high INFD examples compared to MeLIME. We can thus attribute the lower INFD of MeLIME to possible overfitting especially given that their GI are comparable and, stability and unidirectionality are worse for MeLIME.
>
> 2) **uses them to make decisions:** We have now weakened statement to say "might rely on them to make decisions".
>
> 3) **Assumption 1:** We have now (rephrased) clarified this in the paper. Essentially, we assume each of the dimensions to be independent, which can happen when we add independent noise as is seen with random neighborhood generation in LIME type methods.
>
> 4) **Assumption 2 implication:** The assumption states that the $l_{\infty}$ penalty implies the $l_1$ penalty, and hence in our analysis the behavior of the method will only be constrained by it.
>
> 5) **Title:** Causality (Pearl, 2009) typically tries to identify factors that are part of a global data generating process. In the situation of local explainability we care about factors that the black-box model utilizes to make a local decision. Hence, all globally causal factors may not be relevant to a locality. For example, consider a black-box model $f$ that bases its decisions on two features $x_1, x_2$
> in one locality and only on $x_1$ in the other. More precisely, let $f=g_1(x_1,x_2)$ and $f=g_2(x_1)$ be the models in each locality respectively, where $g_1$ and $g_2$ are the local data generators/functions. Globally, both features $x_1$ and $x_2$ are causal to the model. However, from a local standpoint in the second locality only $x_1$ is causal. It is possible because of random neighborhood generation and other instabilities that LIME type of methods might incorrectly pick up both $x_1$ and $x_2$ as the relevant features in this locality. However, given the conservative behavior of our approach described in section 4.3 our method is more likely to pick just $x_1$ which presumably has a more persistent correlation with the output of the model. This stability is witnessed in the experiments.
> This is also the reason why we titled our paper "...Towards Causal..." as we are more likely to uncover the true local factors in a local data generating DAG, nonetheless were careful to refrain from saying point blank "...Causal explanations..." as that we agree would be a much bolder claim. As such, if you still believe the title is somewhat misleading we would be open to replacing the word "Causal" with say "Robust" or "Robust and Intuitive" or "Stable and Unidirectional" or some other suggestion you might have.
>
> 6) **Minor Comments:** We have fixed typos and added the citation suggested by you.

---

### Official Review · Reviewer_H918 · 2021-11-08

**Correctness:** 3
**Technical Novelty And Significance:** 4
**Empirical Novelty And Significance:** 2
**Recommendation:** 5
**Confidence:** 2

**Main Review:**

**Strengths**

The introduction presents a good motivation and explains well the gap in current research. Although you do a great job defining locality later in the paper, a sentence defining what is meant by local decisions and locality would make the intro easier to read. Similar for the terms faithful, stable, and unidirectional, which are technical terms that are used before definition in the intro.

The related work does a great job of summarizing global vs local explainability, exemplar vs feature based methods, as well as robust and causal approaches.

The preliminaries offers a faithful explanation of IRM and does well to distinguish local explainability from IRM.

The description of Nash equilibria and definition of a pure strategy Nash equilibrium as well as the Desirable properties are very well written, but I wonder if this shouldn't be part of the preliminaries? With the exception of unidirectionality, which may be considered a contribution?

The method description well describes and justifies the main contributions of the paper.

The experiments section does well to describe the datasets and metrics used.

**Weaknesses**
The main issues I have are with the experiments section.

1. I'm not sure that Figure 2. demonstrates that the explanations for LINEX are better than MeLIME. They. do look smoother, but does that make them better explanations? I don't like the use of the correlation coefficient here to imply that the LINEX explanations are better as I could just put the original image as the explanation and have a perfect score. For the shoe, why is the LINEX explanation better than MeLIME?

2. There doesn't seem to be a great deal of difference in the top attributed works for the sentiment analysis experiment. Is it possible to look at the top attributed bi-grams rather than just the top attributed words?

3. I appreciate the effort to report standard errors, but I'm not sure it makes sense to take these over different kernel sizes. How is this a sensible approach? Assuming that this is ok, why highlight 1% improvements rather than statistical significance?

**Summary Of The Paper:**

This paper proposes an extension of Locally interpretable model agnostic explanations (LIME) called Locally invariant explanations (LINEX).
LINEX is inspired by invariant risk minimization and aims to provide high-fidelity, stable, black-box invariant, unidirectional local explanations.
LINEX is empirically evaluated using tabular, image, and text data and the authors show improvement over LIME and other baselines over several metrics and datasets.

**Summary Of The Review:**

I think this is an interesting paper that has technical novelty, however there are issues with the experimental results that need to be addressed.

---

> ### Author Response · Authors · 2021-11-16
> **Response to reviewer H918**
>
> 1) **Clarifying local decisions, faithfulness etc. in the introduction:** We have now indicated what local decisions, an explanation being faithful, stable and unidirectional mean in the introduction by describing them in few words. For example, for local decisions we say this means example specific decisions etc. Given the space constraints we kept these descriptions concise. However, if you believe this has to be more elaborately brought out we would be happy to do so.
>
> 2) **Move NE and desirable properties to preliminaries:** Thanks for the suggestion. We have now moved the NE description to preliminaries. However, we kept the Desirable Properties subsection as it is since, we did not want to break Unidirectionality from the rest. Nevertheless, we now mention clearly at the beginning of the section that this is our contribution and the remaining three properties are from prior works.
>
> 3) **Why Fig. 2 supports LINEX?** You are correct in asserting that replicating the original image will lead to a maximum $r$ value. However, the explanations shown in Fig. 2 are for a similar level of sparsity for both MeLIME and LINEX, and hence the fact that LINEX recovers the shape of the sandal which seems to be an important in its recognition is significant. Fig. 20 in the appendix shows how the mean attributions using the two methods for a particular class correlate with the mean pixel values of all images belonging to that class. This evaluation makes sense for FMNIST because images in a class are well registered (i.e. similar location and orientation). We could move a row from that figure replacing the one in Fig. 2 if you think it conveys our message better.
>
> 4) **Why average over kernel widths?** Many a times identifying the precise kernel width to use is not straightforward (Zhang et al. 2019b). Hence, we averaged over multiple of these (rather than cherry picking one) to test the robustness of the different explainability methods over a range of such choices, as ideally one would want some level of stability w.r.t. them for the method to be trustworthy in practice.
>
> 5) **Why 1\% improvement is highlighted?** Given the small standard errors we felt 1\% improvement was a more stringent criterion. We now have added Table 5 in the appendix which highlights statistically significant results and as we can see (qualitatively) the insights are unchanged.
>
> 6) **Why not top-attributed bi-grams?** The vocabulary size was 15670 which would have been in the 100s of millions if we looked at bi-grams instead. This is the reason for sticking to word attributions.

---

> > ### Author Response · Authors · 2021-11-30
> > **Checking in**
> >
> > We believe we have addressed many of your concerns. Please let us know if you have any more questions. Thank you.

---

### Author Response · Authors · 2021-11-16
**Common Response**

We thank all the reviewers for their constructive comments. Based on the reviews we have made the following changes:

1) Added few word descriptions for what local decisions, faithfulness, stability and unidirectionality mean in the introduction.

2) Moved NE definition to Preliminaries section.

3) Added Table 5 in the appendix indicating statistically significant results in addition to 1\% improvement results that are already highlighted in the main paper.

4) Added Section G to the appendix showing examples of LINEX vs MeLIME where LINEX INFD is the highest (error analysis).

5) Commented on run times for MeLIME and MAPLE in Section A of the appendix.

6) Addressed other minor comments such as typos, rephrasing and adding citations.

We now address each of your concerns individually.

---

### Author Response · Authors · 2021-12-06
**Checking in...**

Thanks to all reviewers for their critical reviews. Additionally, thanks to reviewers tbk2 and 388N for the engaging discussions. We believe we have addressed most of the reviewers concerns. Please let us know if any more clarifications or explanations are required. Thank you.

---

### Decision · Program_Chairs · 2022-01-20

**Decision:**

Reject

**Comment:**

The reviewers are largely in agreement that this proposal would benefit from more clarity and comparison to key papers/findings in this space. While one reviewer is leaning towards acceptance, and their points were considered by the other reviewers, there wasn't a consensus towards aligning towardsa an acceptance. Thus, I recommend that the authors take advantage of the reviewers' comments to further improve their manuscript.